# Align Your Query: Representation Alignment for Multimodality Medical Object Detection

## Abstract

Medical object detection suffers when a single detector is trained on mixed medical modalities (*e.g.*, CXR, CT, MRI) due to heterogeneous statistics and disjoint representation spaces. To address this challenge, we turn to representation alignment, an approach that has proven effective for bringing features from different sources into a shared space. Specifically, we target the representations of DETR-style object queries and propose a simple, detector-agnostic framework to align them with modality context. First, we define *modality tokens*: compact, text-derived embeddings encoding imaging modality that are lightweight and require no extra annotations. We integrate the modality tokens into the detection process via *Multimodality Context Attention* (MoCA), mixing object-query representations via self-attention to propagate modality context within the query set. This preserves DETR-style architectures and adds negligible latency while injecting modality cues into object queries. We further introduce *QueryREPA*, a short pretraining stage that aligns query representations to their modality tokens using a task-specific contrastive objective with modality-balanced batches. Together, MoCA and QueryREPA produce modality-aware, class-faithful queries that transfer effectively to downstream training. Across diverse modalities trained altogether, the proposed approach consistently improves AP with minimal overhead and no architectural modifications, offering a practical path toward robust multimodality medical object detection.

## 1 Introduction

Medical object detection, the task of identifying and localizing specific anatomical structures or abnormalities within medical images, is a cornerstone of modern computer-aided diagnosis and clinical decision support systems. Conventional object detection methods originally developed for natural images (Sun et al., 2021; Zhu et al., 2020; Zhang et al., 2022a; Li et al., 2022; Chen et al., 2023; Liu et al., 2024) achieve strong results in the medical domain when applied within a single imaging modality. However the prevailing trend in foundation models is towards the integration of diverse data types, and the biomedical domain is no exception (Moor et al., 2023; Tu et al., 2024). This is a paradigm that introduces significant complexity in the biomedical field, which is characterized by a wide array of imaging modalities such as X-ray, CT, MRI, colonoscopy, etc.

This inherent heterogeneity in medical data of different imaging modalities often leads to degraded performance when a single model is trained on a mixed multimodality[1] dataset (Guan & Liu, 2021; Kang et al., 2023; Yang et al., 2024). The distinct statistical properties and visual characteristics of each imaging type create a complex, disjoint representation space (Cho et al., 2025). Consequently, developing a model that performs robustly across these varied modalities is a formidable challenge. The central problem, therefore, is the generation of efficient and effective representations that can generalize across a mixed corpus of highly diverse medical imaging data.

To address this, we turn to *representation alignment*, a technique that has recently gained significant traction for its ability to harmonize representations from disparate sources (Wang & Isola, 2020). Recently popularized in generative models through methods like REPA (Yu et al., 2024), representation alignment seeks to bridge the semantic gap between different modalities, thereby creating a

---

[1]*Multimodality* defined to be the state of containing multiple imaging modalities (*e.g.*, CXR, CT, MRI); this is not to be confused with *multimodal*, the state of containing multiple data types (*e.g.*, text, image).

more cohesive and informative shared representation space (Lee et al., 2025; Yoon et al., 2025). We posit that this concept can be powerfully repurposed for the task of medical object detection in multimodality datasets. In this work, we propose to leverage the core principles of representation alignment to enhance the quality of representations for multimodality medical object detection.

Specifically, we introduce a novel approach that redefines the role of representation alignment by focusing on *object queries*, the learnable embeddings that directly inform class prediction and bounding box regression in modern DETR-like detection architectures (Carion et al., 2020). We align the object queries with *modality tokens*, which serve as text-derived representations that encode each imaging modality (*e.g.*, CXR, CT, MRI, endoscopy) and target class. The modality tokens act as stable anchors shared across the mixed multimodality dataset with favorable properties: they are lightweight to generate, require no extra annotations, and make modality context explicit without altering backbone or head designs.

For integration of modality tokens, we propose **M**ultim**o**dality **C**ontext **A**ttention (**MoCA**), a novel self-attention mechanism designed to mix the representations of object queries. Rather than adding a separate language stream, MoCA simply appends the relevant modality token to the detector's query set and lets the decoder's self-attention mix the token with object queries. This keeps the architecture faithful to DETR-style designs, adds negligible latency, and injects modality cues within the evolving query representations precisely where decisions are formed. Inspired by recent methods for multimodal integration in generative models (Esser et al., 2024), MoCA is, to the best of our knowledge, the first method to explore *intra-query mixing* of representations for object detection. This represents an important departure from conventional multimodal integration techniques that pre-train the image encoder or utilize cross-attention (Li et al., 2022; Liu et al., 2024), widening our interest to multimodal integration of object queries.

Furthermore, to strengthen the alignment, we introduce a pre-training stage where object query representations are aligned with modality tokens guided by a contrastive **Query Representation Alignment** (**QueryREPA**) loss. Such alignment occurs before any detection head is trained, shaping a query manifold that is both modality-aware and class-faithful. To facilitate this process and prevent any single modality from dominating the training, we perform *modality batch sampling* to ensure each batch contains a balanced mix of images from different modalities. Our experiments demonstrate that QueryREPA and MoCA are complementary, yielding a synergistic effect that significantly boosts medical object detection performance in mixed multimodality datasets.

In summary, our main contributions are:

- A simple, detector-agnostic formulation of modality tokens that makes modality and class context explicit with minimal overhead. These modality tokens are lightweight to generate and require no extra annotations.

- Multimodality Context Attention (MoCA), a drop-in self-attention mechanism that appends modality tokens to queries, strengthening multimodality detection without architectural modifications.

- QueryREPA, a pretraining stage that aligns query representations to modality tokens using a task-specific contrastive loss. Combined with MoCA, the pretraining effectively boosts downstream AP across diverse medical imaging modalities.

## 2 RELATED WORK

### 2.1 MULTIMODALITY IN THE MEDICAL DOMAIN

The rapid push toward generalist biomedical AI has intensified interest in training models across heterogeneous imaging modalities (e.g., CXR, CT, MRI, pathology), but domain shifts and modality-specific statistics often degrade performance in unified settings (Moor et al., 2023; Tu et al., 2024). Works on medical domain adaptation and cross-modality transfer further highlight the difficulty of learning modality-robust features (Guan & Liu, 2021; Kang et al., 2023; Yang et al., 2024). In parallel, contrastive pretraining with paired medical image–text data (e.g., ConVIRT (Zhang et al., 2022b)) demonstrates that textual supervision can structure representation spaces with clinically meaningful semantics, motivating tokenized, text-derived signals as efficient context carriers for vision models. More recently, M$^3$Bind (Liu et al., 2025) proposes a pretraining framework that uses

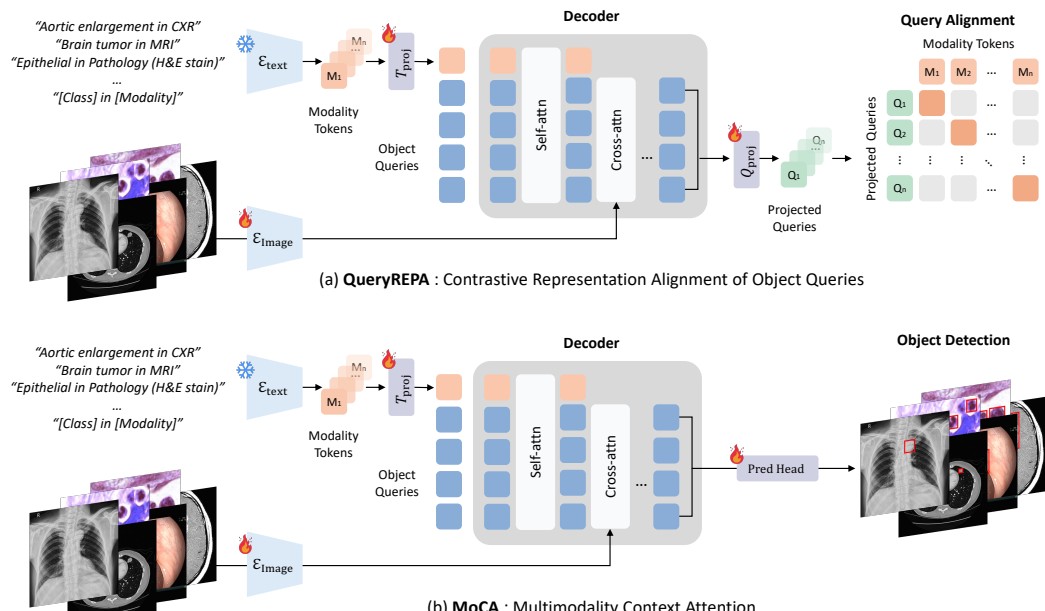

Figure 1: **(a) Contrastive Representation Alignment of Object Queries (QueryREPA):** During a pretraining stage, each image in a modality-balanced batch is associated with a text-derived *modality token*. For each image, we mean-pool the decoder's object-query embeddings into a single query representation, project it into the modality-token space, and optimize a contrastive alignment loss between the projected queries and the modality tokens, yielding a modality-aware query manifold. **(b) Multimodality Context Attention (MoCA):** For a given image, a modality token is selected and concatenated with the set of object queries to form an augmented query set. Information fusion occurs in the self-attention layer of the decoder, allowing each object query to attend to the modality token to explicitly attain modality-specific context.

a shared text embedding space to bind diverse medical imaging modalities (*e.g.*, X-ray, CT, retina, ECG, pathology) without requiring explicit cross-modality image pairs. Our formulation builds on these trends by introducing compact, text-derived modality tokens that make modality context explicit and reusable across detectors and datasets.

## 2.2 OBJECT DETECTION

Transformer-based detectors first introduced in DETR (Carion et al., 2020) have reshaped end-to-end detection through query-based decoding and set prediction, with subsequent advances improving optimization (*i.e.*, Deformable DETR (Zhu et al., 2020)) and feature sampling (*i.e.*, DINO (Zhang et al., 2022a)). Alternative paradigms such as Sparse R-CNN (Sun et al., 2021) and diffusion-based detectors (Chen et al., 2023) explore different proposal and training dynamics. For language-guided open-set detection, GLIP (Li et al., 2022) and Grounding DINO (Liu et al., 2024) blend text with vision, typically by threading text tokens via cross-attention into the decoder. In the medical setting, recent efforts tailor DETR-style pipelines to clinical imagery and constraints (Ickler et al., 2023). Distinct from cross-attention fusion, our Multimodality Context Attention (MoCA) appends modality tokens directly to the query set and performs self-attention mixing inside the decoder, injecting modality cues precisely at the locus where decisions are formed, while preserving detector architecture and latency characteristics.

## 2.3 REPRESENTATION ALIGNMENT FOR OBJECT DETECTION

Representation alignment has emerged as a practical principle for reconciling heterogeneous embeddings: aligning and uniforming features improves transfer and robustness in discriminative regimes (Wang & Isola, 2020; Duan et al., 2022; Xiao et al., 2024). Existing representation align-

ment methods for *object detection* can be roughly grouped into three paradigms: **(i) backbone-level alignment** of global image features, where contrastive or CLIP-style objectives act only on outputs of the *image encoder* and do not modify the logic of the detection model at all (*e.g.*, MoCo (He et al., 2020), SimCLR (Chen et al., 2020), SwAV (Caron et al., 2020), ConVIRT (Zhang et al., 2022b)); **(ii) region-centric alignment**, which aligns *local* pre-defined region or proposal features to text descriptions (*e.g.*, RegionCLIP (Zhong et al., 2022), DITO (Kim et al., 2024)); and **(iii) query-level alignment (ours)**, which directly aligns object-level embeddings or "queries" to compact modality tokens inside the decoder. Therefore, our framework differs from existing works in both what is being aligned *and* the target of the alignment. Thus, existing representation alignment methods are largely *complementary* to our method: in principle, it would be possible to pretrain the image encoder with backbone-level alignment and afterwards apply our query-level alignment.

## 3 REPRESENTATION ALIGNMENT FOR MEDICAL OBJECT DETECTION

We address multimodality heterogeneity by making modality context explicit inside the query set of DETR-style detectors. First, we construct compact, text-derived **modality tokens** for each (modality, class) pair; these tokens act as stable anchors that expose modality semantics without altering backbones or heads. Next, **Multimodality Context Attention (MoCA)** appends the relevant token to the object-query set and mixes them with queries via decoder self-attention, injecting modality cues with negligible latency (see Figure 1b). Finally, **Query Representation Alignment (QueryREPA)** pretrains the query space by aligning query statistics to their modality tokens using a contrastive objective with modality-balanced batches (see Figure 1a).

### 3.1 MODALITY TOKENS

**Construction of Modality Tokens.** Prior to multimodality representation alignment, we first construct a set of *modality tokens* to serve as an effective set of representations for each imaging modality. Each modality token contains semantic information regarding its class and imaging modality, encouraging disentanglement among imaging modalities in multimodality datasets. Specifically, let $\mathcal{T}_{d,c} = \{[\text{CLS}], w_1, w_2, \cdots, w_L, [\text{SEP}]\}$ be a sequence of words comprising the textual input describing the imaging modality $d \in \mathcal{D}$ (*e.g.*, CXR, CT, MRI, pathology, colonoscopy) and target class of interest $c \in \mathcal{C}$ (*e.g.*, aortic enlargment). A pretrained text encoder $\mathcal{E}$ transforms $\mathcal{T}_{d,c}$ into a set of embeddings:

$$\mathcal{E}(\mathcal{T}_{d,c}) = [\boldsymbol{e}_{[\text{CLS}]}, \boldsymbol{e}_1, \boldsymbol{e}_2, \cdots, \boldsymbol{e}_L, \boldsymbol{e}_{[\text{SEP}]}] \in \mathbb{R}^{d_{\text{text}} \times (L+2)} \quad (1)$$

where $d_{\text{text}}$ is the dimension of the text-encoder output space. In practice, using all $L+2$ tokens would incur an impractical time complexity. Thus, we propose to leverage only the embedding for the $[\text{CLS}]$ token, denoted as $\boldsymbol{e}_{[\text{CLS}]} \in \mathbb{R}^{d_{\text{text}} \times 1}$. The $[\text{CLS}]$ token is mapped by the linear projection $\boldsymbol{W} \in \mathbb{R}^{d_{\text{model}} \times d_{\text{text}}}$ to define the modality token $\boldsymbol{m}_{d,c}$ generated from $\mathcal{T}_{d,c}$:

$$\boldsymbol{m}_{d,c} := \boldsymbol{W}\boldsymbol{e}_{[\text{CLS}]} \in \mathbb{R}^{d_{\text{model}}} \quad (2)$$

We create a modality token $\boldsymbol{m}_{d,c}$ for each pair of imaging modality $d \in \mathcal{D}$ and target class $c \in \mathcal{C}$ to create a fixed set of modality tokens $\mathcal{M}_{\mathcal{E}}$:

$$\mathcal{M}_{\mathcal{E}} := \{\boldsymbol{m}_{d,c} : d \in \mathcal{D}, c \in \mathcal{C}\}. \quad (3)$$

where $\mathcal{E}$ is the pretrained encoder of choice. The constructed set of modality tokens $\mathcal{M}_{\mathcal{E}}$ is an effective set of representations for the imaging modalities and classes in our problem of interest. As a different selection of encoder $\mathcal{E}$ would generate a different set of modality tokens $\mathcal{M}_{\mathcal{E}}$, we explore the effects of using various sets $\mathcal{M}_{\mathcal{E}}$ in our experiments.

### 3.2 MOCA : MULTIMODALITY CONTEXT ATTENTION

To integrate the knowledge of modality tokens $\mathcal{M}_{\mathcal{E}}$, we propose MoCA, a novel method to refine object queries during the detection process via self-attention in the detection decoder. Contrary to prior methods that do not exhibit multimodal integration (Figure 2a) or integrate via cross attention (Figure 2b), we create a comprehensive fused representation space between modality tokens and object queries (Figure 2c). Thus, the modality tokens act as *semantic anchors* or *context providers* to guide the refinement of individual object queries for detection by making them modality-informed.

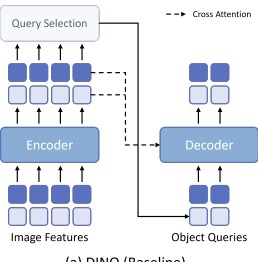 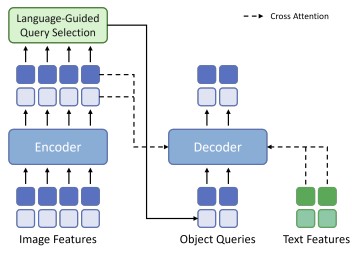 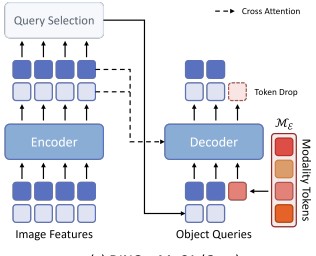

(a) DINO (Baseline)      (b) Grounding DINO      (c) DINO + MoCA (Ours)

Figure 2: Comparison of text/modality integration in DETR-style detectors. **(a) DINO (baseline):** image-only encoder–decoder with query selection. **(b) Grounding DINO:** language-guided query selection and cross-attention in the decoder (object queries attend to a sequence of text tokens). **(c) Ours (DINO+MoCA):** append compact modality tokens to the object queries and fuse by self-attention in the decoder; the tokens act as semantic anchors that refine queries in a modality-aware way with minimal overhead.

Formally, given an input image obtained from imaging modality $d \in \mathcal{D}$ with target class $c \in \mathcal{C}$, we leverage the modality token $\boldsymbol{m}_{d,c} \in \mathcal{M}_\mathcal{E}$ as the target for fusion. The matrix of $N$ object queries $\boldsymbol{Q} = [\boldsymbol{q}_1, \cdots, \boldsymbol{q}_N] \in \mathbb{R}^{N \times d_{model}}$ is concatenated with the modality token $\boldsymbol{m}_{d,c}$ through a learnable linear projection $f_\theta$ to form an augmented query set $\tilde{\boldsymbol{Q}}$ as follows:

$$\tilde{\boldsymbol{Q}} = [\boldsymbol{q}_1, \cdots, \boldsymbol{q}_N, f_\theta(\boldsymbol{m}_{d,c})] \in \mathbb{R}^{(N+1) \times d_{model}} \tag{4}$$

This augmented set $\tilde{\boldsymbol{Q}} = [\boldsymbol{Q}, f_\theta(\boldsymbol{m}_{d,c})]$ is then fed into a Multi-head Self-Attention (MSA) layer of the detection decoder. The queries, keys, and values for the attention computation are all derived from linear projections of $\tilde{\boldsymbol{Q}}$. Let $\boldsymbol{Q}^{(l)}$ denote the set of queries $\boldsymbol{Q}$ at layer $l$ of the detection decoder. The overall attention operation is thus:

$$\text{MSA}(\tilde{\boldsymbol{Q}}) = \text{softmax}\left(\frac{(\tilde{\boldsymbol{Q}}\boldsymbol{W}_Q)(\tilde{\boldsymbol{Q}}\boldsymbol{W}_K)^T}{\sqrt{d_k}}\right)(\tilde{\boldsymbol{Q}}\boldsymbol{W}_V) \tag{5}$$

$$\boldsymbol{Q}^{(l+1)} = \text{MSA}\left(\tilde{\boldsymbol{Q}}^{(l)}\boldsymbol{W}_Q, \tilde{\boldsymbol{Q}}^{(l)}\boldsymbol{W}_K, \tilde{\boldsymbol{Q}}^{(l)}\boldsymbol{W}_V\right) + \boldsymbol{Q}^{(l)} \tag{6}$$

where $\boldsymbol{W}_Q, \boldsymbol{W}_K, \boldsymbol{W}_V$ are learnable projection matrices. Because $\boldsymbol{Q}^{(l+1)}$ is an explicitly learned function of $(\boldsymbol{Q}^{(l)}, \boldsymbol{m}_{d,c})$, training with the final detection loss encourages the model to *use* the informative pathway from $\boldsymbol{m}_{d,c}$. This increases the statistical dependence between $\boldsymbol{Q}^{(l+1)}$ and $\boldsymbol{m}_{d,c}$, raising the mutual information $I(\boldsymbol{Q}^{(l+1)}; \boldsymbol{m}_{d,c})$ relative to an identical decoder lacking $\boldsymbol{m}_{d,c}$.

The critical consequence of this formulation is that every query in the augmented set can attend to every other query. This means that each of the $N$ object queries can *refer to* information from the modality token. The attention weight between an object query and the modality token determines the degree to which the contextual information encoded in the modality token influences the object query's representation in that layer. This process is repeated across all decoder layers, allowing for iterative refinement of object queries based on the continuous infusion of modality-specific context.

### 3.3 QUERYREPA: CONTRASTIVE REPRESENTATION ALIGNMENT OF OBJECT QUERIES

Modern query-based detectors use an encoder–decoder transformer that maintains, at each decoder layer $l$, a set of object-query embeddings $\{\boldsymbol{q}_i^{(l)}\}_{i=1}^N \in \mathbb{R}^{d_{\text{model}}}$. In standard training, these queries are initialized identically across all imaging modalities and are shaped only indirectly through the detection loss. As a result, they do not explicitly encode modality-specific semantics and often form an entangled representation space across modalities (see Appendix G). To address this limitation, we explicitly align the query embeddings with text-derived modality tokens. Specifically, QueryREPA takes the decoder outputs $\{\boldsymbol{q}_i^{(l)}\}$ at a chosen layer $l$ as the query embeddings and align them with their corresponding modality tokens. We apply QueryREPA only to layers after cross-attention has injected image features, ensuring that the queries we align are informative about the input image.

**Query Representation Alignment.** For the set of object-query embeddings $\{\boldsymbol{q}_i^{(l)}\}_{i=1}^N$, we form a mean query representation $\overline{\boldsymbol{q}}^{(l)} = \frac{1}{N}\sum_{i=1}^N \boldsymbol{q}_i^{(l)}$. This mean-pooled vector aggregates information across all queries for the image and is what we align to the modality token $\boldsymbol{m}_{d,c} \in \mathcal{M}_{\mathcal{E}}$. To facilitate the alignment between query embeddings $\{\boldsymbol{q}_i^{(l)}\}_{i=1}^N$ and modality token $\boldsymbol{m}_{d,c}$, we introduce a simple learnable projection $g_\phi$ that maps the query embeddings into the latent space of $\mathcal{M}_{\mathcal{E}}$. The Query Representation Alignment (**QueryREPA**) loss is defined as:

$$\mathcal{L}_{\text{QRA}}(\overline{\boldsymbol{q}}^{(l)}, \boldsymbol{m}_{d,c}) = -\log \frac{\exp(\text{sim}(g_\phi(\overline{\boldsymbol{q}}^{(l)}), \boldsymbol{m}_{d,c})/\tau)}{\sum_j \exp(\text{sim}(g_\phi(\overline{\boldsymbol{q}}^{(l)}), \boldsymbol{m}_{d,c}^{(j)})/\tau)} \tag{7}$$

where $\text{sim}(\cdot,\cdot)$ denotes cosine similarity and $\tau > 0$ denotes the temperature. This loss is a direct adaptation of the InfoNCE loss (Oord et al., 2018) well established in the context of representation learning. We run QueryREPA as a short pretraining stage before end-to-end detection training: the backbone and decoder are updated using only $\mathcal{L}_{\text{QRA}}$, and the detection heads are not used; afterwards, the full detector is trained with standard detection losses on top of newly initialized weights.

**Modality Batch Sampling.** Let $\mathcal{D}$ denote the set of imaging modalities (e.g., CXR, CT, MRI, endoscopy, pathology), $|\mathcal{D}| = M$. We construct mini-batches $\mathcal{B} = (x_b, d_b, c_b)_{b=1}^B$ such that all samples come from *distinct* modalities:

$$d_b \neq d_{b'} \quad \text{for all } b \neq b', \quad B \leq M. \tag{8}$$

In practice, we implement a round-robin, per-modality queue sampler that (i) cycles uniformly over $\mathcal{D}$ to mitigate modality imbalance, (ii) draws one example per selected modality, and (iii) optionally shuffles classes $c_b$ within each modality-specific queue. This guarantees that, for any anchor image $x$ from modality $d$, the in-batch *negatives* used by the QueryREPA loss of Eq. (7) are tokens from *other* modalities $d' \neq d$. Such batch construction focuses the contrast explicitly on *cross-modality* separability, yielding stronger, well-conditioned gradients for modality-aware alignment. When $B < M$, we randomly sample the subset of covered modalities with uniform probability across iterations so that, amortized over training, each modality pair is contrasted with high probability.

**QueryREPA as MI maximization.** The proposed QueryREPA induces mutual information maximization of object queries. Define $U^{(l)} := g_\phi(\overline{\boldsymbol{q}}^{(l)})$ the projected query statistic at decoder layer $l$ and $V := \boldsymbol{m}_{d,c}$ the modality token. Note that QueryREPA is performed on a layer $l$ *after* image features are inserted via cross-attention (*i.e.*, $l \neq 1$ to ensure queries $\overline{\boldsymbol{q}}^{(l)}$ is informative of the input image $x$). Using the InfoNCE objective on positive pairs $(U^{(l)}, V)$ and $K$ in-batch negatives $\{V_k\}$,

$$\mathcal{L}_{\text{QRA}}^{(l)} = -\mathbb{E}\left[\log \frac{\exp(\text{sim}(U^{(l)}, V)/\tau)}{\exp(\text{sim}(U^{(l)}, V)/\tau) + \sum_{k=1}^K \exp(\text{sim}(U^{(l)}, V_k)/\tau)}\right]. \tag{9}$$

**Remark 1.** *Given the InfoNCE objective of Eq. (9) on positive pairs $(U^{(l)}, V)$ and $K$ in-batch negatives $\{V_k\}$, the standard lower bound is expressed as*

$$I(U^{(l)}; V) \geq \log(1 + K) - \mathcal{L}_{QRA}^{(l)}. \tag{10}$$

Thus, minimizing $\mathcal{L}_{\text{QRA}}$ *maximizes* a variational lower bound on $I\big(g_\phi(\overline{\boldsymbol{q}}^{(l)}); \boldsymbol{m}_{d,c}\big)$. Because our modality batch sampling ensures $V_k$ are tokens from different modalities, the same objective simultaneously decreases the association of the queries with mismatched modalities (i.e., pushes $I\big(U^{(l)}; V_k\big)$ down indirectly via the softmax competition). The bound tightens as $K$ increases, indicating that a larger $B$ (up to $M$) would result in higher performance. This remark restates a standard property of InfoNCE; we include it to clarify its implications under modality-balanced batch sampling in QueryREPA.

## 4 EXPERIMENTS

### 4.1 EXPERIMENTAL SETTINGS

**Datasets.** We create a *single mixed multimodality dataset* via aggregation of multiple sub-datasets. The sub-datasets we use vary in terms of image resolution, number of annotations, and class distributions, to ensure a robust evaluation across different imaging modalities. Specifically, we use

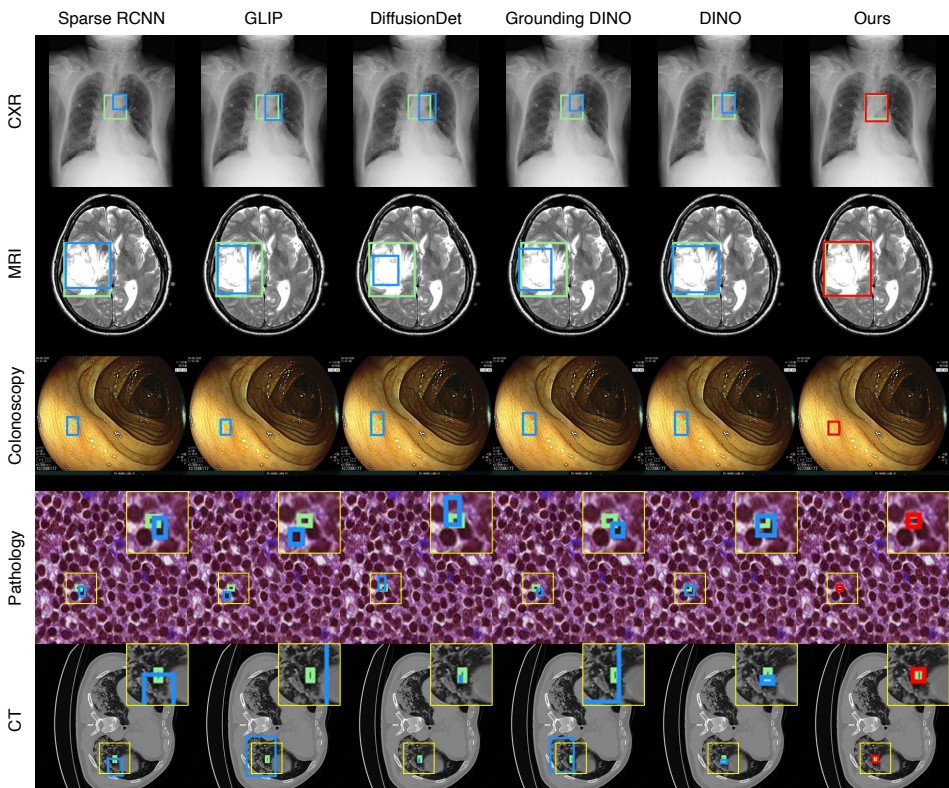

Figure 3: **Qualitative Comparison.** Comparison results between various state-of-the-art detection methods and the proposed method is shown above. Our method effectively leverages modality context to significantly enhance anomaly localization (highlighted in red), compared to baseline results (highlighted in blue). Ground truth bounding boxes are highlighted in green. For cases where the bounding boxes are small, insets show an enlarged view of the highlighted yellow region.

VinBigData Chest X-ray (VinBigData, 2021) for CXR; COVID-19 CT (MVD, 2020) and LIDC-IDRI (Consortium, 2008) for CT; NeoPolyp (BKAI-IGH, 2021) for colonoscopy; BR35H brain tumor (Hamada, 2021) and ACDC cardiac MRI (Challenge, 2017) for MRI; MoNuSeg (Kumar et al., 2019) for pathology. Detailed description regarding the datasets used is provided in Appendix C.

**Training Details.** We train all models on a single RTX 3090/4090 GPU and report results from the checkpoint achieving the highest validation AP. Models are optimized using AdamW with a base learning rate of $1 \times 10^{-4}$ (Deformable DETR) or $2 \times 10^{-4}$ (DINO) and a MultiStep learning-rate decay scheduled at epoch 40 and epoch 30, respectively, for a total of 50 and 36 epochs. We adopt standard DETR-style multi-scale data augmentation. Detailed configurations, augmentation schedules, and hyperparameters are provided in Appendix D for reproducibility.

Regarding model usage, we assess the impact of leveraging various encoders for the construction of modality tokens. Specifically, we use CLIP (Radford et al., 2021), BiomedCLIP (Zhang et al., 2023), and PubMedCLIP (Eslami et al., 2023) in our experiments. BiomedCLIP and PubMedCLIP are variants of CLIP, tuned specifically for the medical domain. For the image encoder, all experiments are conducted using the ResNet-50 image encoder across all baseline detectors to ensure a fair comparison. For baseline models that utilize self-attention layers requiring positional encoding, such as Deformable DETR and DINO, we apply zero padding to maintain compatibility with architecture.

**Evaluation metrics.** We evaluate using the standard Average Precision (AP) metric (averaged over multiple IoU thresholds) and $AP_{50}$ (at an IoU threshold of 0.5). For comparison with state-of-the-art detection methods, we additionally provide metrics $AP_{75}$, $AP_s$, $AP_m$, $AP_l$. These metrics complement each other by balancing overall detection quality (AP) with more relaxed criterions ($AP_{50}$) to account for varying imaging characteristics, annotation quality, and object scales in medical datasets.

Table 1: Quantitative comparison with state-of-the-art methods in medical object detection. Applying [our method] (MoCA + QueryREPA) on the DINO (Zhang et al., 2022a) detector outperforms all other detection methods. **Bold** = best per column. Underline = second best per column.

| Method | AP | $AP_{50}$ | $AP_{75}$ | $AP_s$ | $AP_m$ | $AP_l$ |
|---|---|---|---|---|---|---|
| Deformable DETR (Zhu et al., 2020) | 35.0 | 54.4 | 37.8 | 17.4 | 27.6 | 38.2 |
| Sparse RCNN (Sun et al., 2021) | 35.9 | 54.4 | 38.8 | 18.4 | 29.0 | 38.0 |
| DiffusionDet (Chen et al., 2023) | 38.2 | 60.3 | 40.1 | 22.4 | 30.6 | 41.6 |
| GLIP (Li et al., 2022) | 37.4 | 57.5 | 39.9 | 20.8 | 30.5 | 42.2 |
| GLIP-A (Li et al., 2022) | 37.7 | 58.3 | 40.1 | 20.9 | 31.0 | 41.4 |
| GLIP-B (Li et al., 2022) | 37.1 | 57.4 | 39.4 | 22.4 | 29.7 | 40.8 |
| GLIP-C (Li et al., 2022) | 37.4 | 57.5 | 40.3 | 21.3 | 29.9 | 42.0 |
| Grounding DINO (Liu et al., 2024) | 32.7 | 50.9 | 35.1 | 17.9 | 26.2 | 33.9 |
| RTMDet (Lyu et al., 2022) | 38.5 | 57.4 | 41.9 | 22.4 | 32.0 | 42.5 |
| Co-DETR (Zong et al., 2023) | 40.2 | 59.1 | **43.6** | 25.9 | 32.7 | 43.3 |
| DINO (Zhang et al., 2022a) | 37.7 | 58.6 | 37.8 | 19.9 | 29.8 | 41.7 |
| DINO + Ours ($\mathcal{E}$: CLIP) | 41.1 | **65.5** | 43.4 | 25.4 | **34.6** | 44.9 |
| DINO + Ours ($\mathcal{E}$: BiomedCLIP) | 40.6 | 65.4 | 42.4 | 24.2 | 33.5 | 44.8 |
| DINO + Ours ($\mathcal{E}$: PubMedCLIP) | **41.3** | **65.5** | 43.3 | **26.6** | 33.2 | **45.4** |

## 4.2 COMPARISON RESULTS

**Qualitative Results.** Figure 3 presents qualitative comparison between various detection methods and our proposed framework across multiple modalities. Baseline detectors frequently miss subtle lesions or produce overly large bounding boxes, resulting in inaccurate localization and blurred lesion boundaries. Our approach enforces modality-invariant semantics and improves the precision of bounding box placement to successfully recover missed findings and produce sharper, clinically meaningful boxes. Additional qualitative examples are included in Appendix H.

**Quantitative Results.** We first provide thorough comparison between our method (*i.e.*, DINO + MoCA + QueryREPA) against a comprehensive suite of state-of-the-art (SOTA) object detectors in Table 1. We integrate modality tokens generated from various powerful text encoders: the general-purpose CLIP, and the domain-specific BiomedCLIP and PubMedCLIP. Our method establishes a new SOTA on this challenging mixed multimodality dataset, outperforming all other models across every metric. With PubMedCLIP, our method achieves a total AP of 41.3 and $AP_{50}$ of 65.5, which are significant improvements over baselines. The performance gains are particularly notable when compared to its own baseline, DINO (37.7 AP), demonstrating the substantial value added by our framework. Furthermore, our approach shows superior performance across different object scales, with leading scores for small ($AP_s$), medium ($AP_m$), and large ($AP_l$) objects. The substantial margin over other methods, including language-guided detectors like GLIP and Grounding DINO, underscores the efficacy of our targeted modality-aware query alignment and attention mechanism for handling the unique challenges of heterogeneous medical data.

We also thoroughly analyze the robustness of our method in Table 2 by observing results for Deformable DETR and DINO. Our framework provides consistent performance gains regardless of the encoder used. For Deformable DETR, our method improves the total AP from a baseline of 35.0 to 37.2 (+2.2) with CLIP-based tokens. For the stronger DINO baseline, the improvement is even more pronounced, boosting the AP from 37.7 to 41.1 (+3.4) and $AP_{50}$ from 58.6 to 65.5 (+6.9). Notably, all three encoders prove effective, confirming that the architectural contributions of QueryREPA and MoCA are the primary drivers of the performance uplift, rather than a dependency on a specific text encoder. This highlights the versatility and robustness of our approach.

**Comparison of using different $\mathcal{M}_\mathcal{E}$.** The robustness of our method allows us to use varying sets of modality tokens $\mathcal{M}_\mathcal{E}$ by differing the type of encoder $\mathcal{E}$ used. Our quantitative comparisons in Table 1 and Table 2 contain results for varying $\mathcal{M}_\mathcal{E}$ by selecting $\mathcal{E}$ as either CLIP, BiomedCLIP, or PubMedClip. Though our framework provides consistent performance increase regardless of $\mathcal{E}$, we find CLIP and PubMedCLIP to consistently perform better than BiomedCLIP. We attribute this to the better disentanglement of modality tokens in CLIP and PubMedCLIP as compared to

Table 2: Quantitative comparison leveraging various modality token encoders $\mathcal{E}$ as CLIP (Radford et al., 2021), BiomedCLIP (Zhang et al., 2023), and PubMedCLIP (Eslami et al., 2023) under the same training setup. Our method shows improved results regardless of $\mathcal{E}$ used, proving its robustness and wide applicability. **Bold** indicates best results per column.

| Method | $\mathcal{E}$ | $\mathcal{L}_{QRA}$ | Total AP | Total AP$_{50}$ | CXR AP | CXR AP$_{50}$ | MRI AP | MRI AP$_{50}$ | Colonoscopy AP | Colonoscopy AP$_{50}$ | Pathology AP | Pathology AP$_{50}$ | CT AP | CT AP$_{50}$ |
|---|---|---|---|---|---|---|---|---|---|---|---|---|---|---|
| Deformable DETR | ✗ | ✗ | 35.0 | 54.4 | 15.1 | 30.9 | 75.8 | 95.9 | 61.3 | 77.3 | 46.5 | 74.1 | 30.5 | 62.3 |
| | CLIP | ✗ | **37.3** | **60.2** | 18.0 | 39.1 | 75.3 | 96.2 | 64.6 | 83.7 | **49.3** | **79.1** | **31.5** | 62.5 |
| | CLIP | ✓ | 37.2 | 60.2 | **18.2** | **39.9** | 76.1 | **96.7** | **66.7** | 84.6 | 45.4 | 76.0 | 31.3 | 61.6 |
| | BiomedCLIP | ✗ | 36.7 | 60.2 | 17.5 | 39.1 | 74.5 | **96.7** | 65.1 | **85.4** | 47.7 | 77.6 | 29.9 | 61.8 |
| | BiomedCLIP | ✓ | 36.7 | 59.6 | 17.7 | 38.9 | 75.7 | 96.5 | 64.2 | 83.4 | 47.1 | 76.8 | 30.1 | 61.0 |
| | PubMedCLIP | ✗ | 36.6 | 59.7 | 17.8 | 39.0 | 75.5 | 96.4 | 64.0 | 84.4 | 46.3 | 76.1 | 29.8 | 61.0 |
| | PubMedCLIP | ✓ | 36.8 | 60.1 | 17.8 | 39.5 | 75.2 | 96.4 | 64.5 | 82.9 | 47.1 | 77.1 | 30.7 | **63.4** |
| DINO | ✗ | ✗ | 37.7 | 58.6 | 17.8 | 36.1 | 78.0 | 96.6 | 65.1 | 81.3 | 50.5 | 77.8 | 33.0 | 66.6 |
| | CLIP | ✗ | 40.8 | 65.1 | 21.1 | 45.3 | 77.0 | 96.9 | 70.4 | 87.2 | 55.5 | 85.1 | **35.1** | 67.2 |
| | CLIP | ✓ | 41.1 | 65.5 | 21.1 | 45.4 | **78.6** | **97.4** | 71.2 | 88.7 | **55.5** | **85.4** | 34.4 | **67.5** |
| | BiomedCLIP | ✗ | 40.9 | **65.7** | 21.2 | 46.3 | 77.9 | 97.2 | 71.3 | 88.0 | 55.1 | 84.7 | 34.2 | 66.9 |
| | BiomedCLIP | ✓ | 40.6 | 65.4 | 21.1 | 45.9 | 77.6 | 97.2 | 70.1 | 85.9 | 54.7 | 85.2 | 33.9 | 67.4 |
| | PubMedCLIP | ✗ | 41.1 | 65.5 | 21.4 | 46.1 | 78.4 | 97.0 | **72.7** | **89.2** | 53.6 | 83.4 | 35.4 | 67.3 |
| | PubMedCLIP | ✓ | **41.3** | 65.5 | **22.0** | **46.5** | 78.1 | 97.0 | 72.4 | 86.9 | 54.1 | 84.0 | 33.7 | 66.6 |

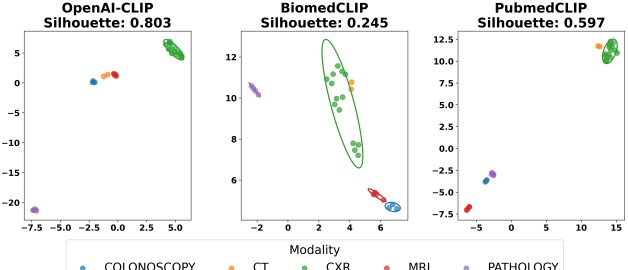

Figure 4: UMAP embedding of modality tokens for varying $\mathcal{E}$.

Table 3: Comparison of decoder head latency (ms) of baseline models with and without MoCA.

| Model | Baseline (ms) | Ours (ms) |
|---|---|---|
| Def. DETR | 7.80 | 8.71 |
| DINO | 9.37 | 9.59 |

BiomedCLIP (visualized in Figure 4). Computation of silhouette scores [2] supports this, showing that better disentanglement between modalities leads to improved performance.

**Runtime Comparison.** The proposed MoCA induces minimal increase in inference time as it incorporates only a single extra token for self-attention. Specifically, we compare decoder head latency (ms) for the two baseline models (*i.e.*, Deformable DETR, DINO) used in Table 3. The reported values are measurements of the average time required per image, measured over 100 samples.

## 4.3 QUERYREPA PRETRAINING ON LARGE-SCALE UNLABELED DATASETS

QueryREPA offers a practical advantage in real-world medical settings where bounding box annotations are scarce. QueryREPA performs representation alignment at the query level, requiring only image-level class labels and *no bounding-box supervision during the pretraining stage*. This makes it particularly suitable for large medical datasets in which detailed annotations are limited.

To demonstrate the potential of QueryREPA, we show below how pretraining on extra images without bounding box labels can increase the final detection performance. As an example, we additionally leverage the widely used NIH Chest X-ray dataset Wang et al. (2017), which contains over 112,000 images but provides bounding-box annotations for fewer than 1,000 cases. This leaves the vast majority of images unusable for conventional detection pretraining. In Table 4, we compare between different datasets used for QueryREPA, while fixing the datasets used for detection training.

---

[2] A metric that evaluates the quality of a clustering solution assessing: 1) how well data points are grouped inside each of their respective clusters, and 2) how well those clusters are separated from each other.

Table 4: Effectiveness of QueryREPA in utilization of unlabeled datasets. Datasets are: (1) NIH with only class labels (2) NIH with bounding box labels (3) VinBigData with bounding box labels. **Bold** = best per column.

| QueryREPA Dataset | Detection Training Dataset | Total AP | Total AP$_{50}$ | CXR AP | CXR AP$_{50}$ | MRI AP | MRI AP$_{50}$ | Colonoscopy AP | Colonoscopy AP$_{50}$ | Pathology AP | Pathology AP$_{50}$ | CT AP | CT AP$_{50}$ |
|---|---|---|---|---|---|---|---|---|---|---|---|---|---|
| ✗ | (2) + (3) | 38.5 | 62.0 | 19.1 | 41.5 | 77.4 | 96.8 | 68.7 | 86.8 | 53.4 | 83.0 | 34.2 | 66.2 |
| (2) + (3) | (2) + (3) | 39.1 | 62.9 | 19.2 | 42.5 | 77.6 | 96.6 | 70.4 | 85.7 | **55.7** | **85.9** | **34.6** | 67.7 |
| (1) + (2) + (3) | (2) + (3) | **39.4** | **63.3** | **19.6** | **42.6** | **78.0** | **97.1** | **72.6** | **89.9** | 54.7 | 84.8 | 34.1 | **68.0** |

Table 5: Ablation of individual components using DINO (Zhang et al., 2022a) as the base model and CLIP (Radford et al., 2021) as the encoder $\mathcal{E}$. Checkmarks denote which modules are enabled. **Bold** = best per column.

| QueryREPA | MoCA | AP | AP$_{50}$ | AP$_{75}$ | AP$_s$ | AP$_m$ | AP$_l$ |
|---|---|---|---|---|---|---|---|
| | | 37.7 | 58.6 | 37.8 | 19.9 | 29.8 | 41.7 |
| ✓ | | 38.1 | 58.4 | 40.9 | 21.0 | 29.8 | 42.3 |
| | ✓ | 40.8 | 65.1 | **43.4** | 25.1 | 33.5 | **45.2** |
| ✓ | ✓ | **41.1** | **65.5** | **43.4** | **25.4** | **34.6** | 44.9 |

We observe that including images without bounding-boxes during QueryREPA pretraining provides measurable performance gains. Thus, QueryREPA *mitigates annotation scarcity* by turning otherwise underutilized, weakly labeled medical data into a beneficial resource that improves detection performance. This provides an entirely new research avenue for performing weakly supervised or unsupervised learning in detection models *directly on object queries*.

### 4.4 ABLATION STUDY

**Ablation of individual components.** A thorough ablation study of individual components (*i.e.*, QueryREPA, MoCA) is provided in Table 5. These experiments confirm that both MoCA and QueryREPA independently contribute to the performance improvements, and that their combination yields the best results due to their synergistic effect.

**Ablation of decoder layer $l$.** We further investigate the effect of applying QueryREPA on different decoder layers of the detection head. As shown in Table 6 of the Appendix, applying QueryREPA on Layer 5 shows highest overall AP, but difference is minimal between selection of layers. Thus, QueryREPA is capable of producing optimal results regardless of decoder layer $l$.

## 5 CONCLUSION

In this work, we addressed the challenge of performance degradation in medical object detection models trained on mixed multimodality datasets. Specifically, we demonstrated representation alignment of object queries to effectively manage the inherent statistical heterogeneity across different imaging modalities. Our proposed framework introduces lightweight, text-derived modality tokens to provide explicit cues to the queries. The Multimodality Context Attention (MoCA) mechanism seamlessly integrates these tokens into the detection process, while the QueryREPA pretraining stage creates a modality-aware query set before downstream training. Our experiments show that the synergistic combination of MoCA and QueryREPA consistently improves detection accuracy for a mixed set of medical imaging modalities, including CXR, CT, MRI, and pathology. Our method offers a detector-agnostic, low-overhead, and practical solution, paving the way for the development of robust and generalizable medical object detection models in real-world clinical settings.

**Limitations.** Our method is evaluated only on public datasets, thus additional validation on multi-center clinical data must be further explored. Furthermore, QueryREPA adds pretraining before end-to-end detection training; future work could explore joint alignment training. Despite these limitations, our method remains a practical step toward robust multimodality medical object detection.

ETHICS STATEMENT

We adhere to the ICLR Code of Ethics. Our study uses only publicly available medical imaging datasets (VinBigData, COVID-19 CT, LIDC-IDRI, NeoPolyp, BR35H, ACDC, MoNuSeg) containing no personally identifiable information. No new human-subject data were collected, and thus no IRB approval was required. As our approach (MoCA + QueryREPA) modifies only query representations and introduces a short pretraining stage without altering the backbone or dataset, its safety and fairness properties are largely inherited from the underlying detectors. Potential bias amplification from public datasets is mitigated through modality-balanced sampling (Section 3.3) and per-modality performance reporting (Table 2).

REPRODUCIBILITY STATEMENT

We provide detailed information to enable full reproducibility. Our algorithmic formulation is described in Section 3, with modality token construction and QueryREPA loss given in Eqs. 1–8. Dataset details and annotation procedures are summarized in Appendix C, and full training configurations, augmentation pipelines, and hyperparameters are reported in Appendix D.

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

# A PROOFS

**Remark 1.** *Given the InfoNCE objective of Eq. (9) on positive pairs $(U^{(l)}, V)$ and $K$ in-batch negatives $\{V_k\}$, the standard lower bound is expressed as*

$$I(U^{(l)}; V) \geq \log(1 + K) - \mathcal{L}_{QRA}^{(l)}. \tag{10}$$

*Proof.* Let $U^{(l)}$ and $V$ be the positive pair with joint $p(u, v)$ and marginals $p(u), p(v)$. Let $K$ i.i.d. negatives $\{V_k\}_{k=1}^{K} \sim p(v)$ be sampled independently of $U^{(l)}$. Define scores $s(u, v) = \text{sim}(u, v)/\tau$ and $r(u, v) = \exp(s(u, v))$. The InfoNCE objective of Eq. (9) can be expressed as:

$$\mathcal{L}_{\text{QRA}}^{(l)} = -\mathbb{E}\left[ \log \frac{r(U^{(l)}, V)}{r(U^{(l)}, V) + \sum_{k=1}^{K} r(U^{(l)}, V_k)} \right]. \tag{11}$$

Introduce a uniformly random index $J \in \{0, \cdots, K\}$ and construct the candidate set $\{V_0, \cdots, V_K\}$ by placing the positive at slot $J(V_j \sim p(v|U^{(l)}))$ and drawing the remaining $K$ elements i.i.d. from $p(v)$. By Bayes' rule and exchangeability,

$$p(J = j \mid u, v_{0:K}) = \frac{\frac{p(v_j|u)}{p(v_j)}}{\sum_{m=0}^{K} \frac{p(v_m|u)}{p(v_m)}}. \tag{12}$$

The model's softmax over scores,

$$q_\theta(J = j \mid u, v_{0:K}) = \frac{r(u, v_j)}{\sum_{m=0}^{K} r(u, v_m)}. \tag{13}$$

is exactly the classifier used by InfoNCE. The InfoNCE loss is the average cross-entropy between the true posterior $p(\cdot \mid u, v_{0:K})$ and $q_\theta(\cdot \mid u, v_{0:K})$:

$$\begin{aligned}
\mathcal{L}_{\text{QRA}}^{(l)} &= \mathbb{E}\left[ -\log q_\theta\left( J \mid U^{(l)}, V_{0:K} \right) \right] \\
&= \mathbb{E}\left[ H\left( p(\cdot \mid U^{(l)}, V_{0:K}) \right) + \text{KL}(p\|q_\theta) \right] \geq \mathbb{E}\left[ H\left( J \mid U^{(l)}, V_{0:K} \right) \right].
\end{aligned} \tag{14}$$

Generate $Z = (J, V_{0:K})$ from $V$ by adding i.i.d. negatives and randomly placing the positive. This defines a channel $p(z \mid v)$ using only independent noise, so $U^{(l)} \to V \to Z$ is a Markov chain. By the data processing inequality,

$$I\left( U^{(l)}; V \right) \geq I\left( U^{(l)}; Z \right) \geq I\left( U^{(l)}; J \mid V_{0:K} \right). \tag{15}$$

Since $J$ is uniform and independent of $V_{0:K}$ under the random placement,

$$\begin{aligned}
I\left( U^{(l)}; J \mid V_{0:K} \right) &= H(J \mid V_{0:K}) - H(J \mid U^{(l)}, V_{0:K}) \\
&= \log(1 + K) - \mathbb{E}\left[ H\left( J \mid U^{(l)}, V_{0:K} \right) \right].
\end{aligned} \tag{16}$$

Combining Eq. (15) and Eq. (16),

$$I\left( U^{(l)}; V \right) \geq \log(1 + K) - \mathbb{E}\left[ H\left( J \mid U^{(l)}, V_{0:K} \right) \right]. \tag{17}$$

Applying Eq. (14) to Eq. (17) yields

$$I\left( U^{(l)}; V \right) \geq \log(1 + K) - \mathcal{L}_{\text{QRA}}^{(l)}, \tag{18}$$

which is the desired standard lower bound.

$\square$

| Decoder Layer | AP | $AP_{50}$ | $AP_{75}$ | $AP_s$ | $AP_m$ | $AP_l$ |
|---|---|---|---|---|---|---|
| Layer 2 | 41.0 | 65.8 | 43.5 | 24.0 | 33.3 | 45.0 |
| Layer 3 | 41.0 | 65.6 | 43.3 | 27.0 | 32.8 | 45.1 |
| Layer 4 | 41.0 | 65.8 | 43.1 | 25.3 | 33.3 | 45.3 |
| Layer 5 | 41.1 | 65.5 | 43.4 | 25.4 | 34.6 | 44.9 |

Table 6: **Effect of decoder layer index $l$ for QueryREPA loss.** Results show that choosing $l = 5$ produces highest overall AP, but difference is minimal. Thus, our method is capable of producing optimal results robust to the choice of $l$.

| Dataset | Modality | Images | Resolutions | Train (Annotations) | Test (Annotations) | Class |
|---|---|---|---|---|---|---|
| Vinbig | CXR | 4394 | Variable | 3296 (18024) | 1098 (5931) | 14 |
| COVID-19 CT | CT | 1388 | $1024 \times 1024$ | 1187 (3801) | 201 (240) | 1 |
| LIDC-IDRI | CT | 9122 | $1024 \times 1024$ | 7389 (7927) | 1733 (1957) | 1 |
| NeoPolyp | Colonoscopy | 1000 | $1024 \times 1024$ | 800 (1891) | 200 (454) | 3 |
| Br35H | MRI | 701 | Variable | 500 (564) | 201 (240) | 1 |
| ACDC | MRI | 2827 | $1024 \times 1024$ | 1831 (4934) | 996 (2732) | 3 |
| MonuSeg | Pathology | 1414 | $320 \times 320$ | 1017 (31367) | 397 (13749) | 4 |

Table 7: Dataset summary with image and annotation counts.

## B ABLATION STUDY OF DECODER LAYER

Ablation study regarding the decoder layer $l$ used for alignment in QueryREPA is given in Table 6. The table shows that our method is capable of producing optimal results regardless of decoder layer $l$. In our main experiments, we choose layer index $l = 5$, due to its slight increase (+0.1) in total AP.

## C DATASET EXPLANATION

In cases where datasets provided only segmentation masks without explicit bounding boxes, we generated detection annotations by extracting bounding boxes directly from the segmentation masks. Detailed descriptions of each dataset are provided below.

**VinBigData Chest Abnormalities Dataset (CXR).** The VinBigData Chest Abnormalities Dataset Vin-BigData (2021) contains chest X-ray images annotated by multiple radiologists, aimed at developing automated detection systems for chest abnormalities. The dataset has been converted to COCO format and optimized using the Weighted Boxes Fusion (WBF) technique. It includes 14 classes of abnormalities, such as aortic enlargement, cardiomegaly, lung opacity, and pneumothorax. Images in the dataset have varying resolutions and are divided into training and testing collections.

**COVID-19 CT Dataset (CT).** The COVID-19 CT dataset MVD (2020) consists of CT scans containing COVID-19 infections with pixel-level annotations. This dataset is used for segmentation and classification of COVID-19 related lung abnormalities. All images have a resolution of $1024 \times 1024$ pixels and are split into training and testing sets.

**LIDC-IDRI Dataset (CT).** The LIDC-IDRI dataset Consortium (2008) is a large-scale collection of lung CT scans with nodule annotations from four experienced radiologists. It is widely used in lung nodule detection and segmentation research. The dataset follows the COCO format with images of $1024 \times 1024$ resolution for training and testing purposes.

**NeoPolyp Dataset (Colonoscopy).** The NeoPolyp dataset BKAI-IGH (2021) contains colonoscopy images annotated for polyp detection and classification. This dataset provides images of resolution $1024 \times 1024$ and supports training and testing sets, facilitating the development of deep learning models for polyp detection.

- "Aortic enlargement in CXR"
- "Atelectasis in CXR"
- "Calcification in CXR"
- "Cardiomegaly in CXR"
- "Consolidation in CXR"
- "ILD in CXR"
- "Infiltration in CXR"
- "Lung Opacity in CXR"
- "Nodule/Mass in CXR"
- "Other lesion in CXR"
- "Pleural effusion in CXR"
- "Pleural thickening in CXR"
- "Pneumothorax in CXR"
- "Pulmonary fibrosis in CXR"

- "Brain tumor in MRI"
- "Epithelial in Pathology (H&E stain)"
- "Lymphocyte in Pathology (H&E stain)"
- "Neutrophil in Pathology (H&E stain)"
- "Macrophage in Pathology (H&E stain)"
- "Left heart ventricle in cardiac MRI"
- "Myocardium in cardiac MRI"
- "Right heart ventricle in cardiac MRI"
- "COVID-19 infection in lung CT"
- "Nodule in lung CT"
- "Neoplastic polyp in colon endoscope"
- "Polyp in colon endoscope"
- "Non-neoplastic polyp in colon endoscope"

Figure 5: List of 27 categories used in our experiments.

**BR35H Brain Tumor Dataset (MRI).** The BR35H dataset Hamada (2021) consists of brain MRI scans categorized into tumor and non-tumor cases. The dataset is used for brain tumor detection tasks. The images have variable resolutions and are split into training and testing sets.

**ACDC Dataset (MRI).** The ACDC dataset Challenge (2017) provides cardiac MRI images with expert annotations for segmentation and classification of heart structures. It consists of $1024 \times 1024$ resolution images and supports both training and testing configurations.

**MoNuSeg Dataset (Pathology).** The MoNuSeg dataset Kumar et al. (2019) is designed for nuclear segmentation in histopathology images. It includes diverse samples from multiple organs, enabling the development of robust segmentation models. Images are provided at a resolution of $320 \times 320$ pixels and are divided into training and testing sets.

# D TRAINING DETAILS

**Setup for Deformable DETR with MoCA and QueryREPA.** We train Deformable DETR with a ResNet–50 backbone for 50 epochs using AdamW (lr=$1\times10^{-4}$, wd=$1\times10^{-4}$) with a MultiStep learning-rate decay at epoch 40. We use 300 object queries throughout training. Pretraining optimizes only a modality-balanced contrastive loss (QueryREPA), sampling mini-batches with a `ModalityBatchSampler` to ensure balanced modality coverage. Data augmentation follows the standard DETR recipe, including random horizontal flip, multi-scale random resize of the shorter image side within [480, 800] px, and random absolute-range cropping with re-resizing. Finetuning resumes from the final pretraining checkpoint, training with Hungarian matching using focal loss (weight=2.0, $\alpha$=0.25, $\gamma$=2.0), L1 loss (weight=5.0), and GIoU loss (weight=2.0). We use a per-GPU batch size of 4 with the default sampler.

**Setup for DINO with MoCA and QueryREPA.** We train DINO for 36 epochs using AdamW (lr=$2\times10^{-4}$, wd=$1\times10^{-4}$) with a MultiStep learning-rate decay at epoch 30. Each iteration uses 900 object queries and 100 denoising queries. Pretraining optimizes only a modality-balanced contrastive loss (QueryREPA) with a `ModalityBatchSampler` for balanced sampling. The same data augmentation pipeline as above is applied. Finetuning resumes from the final pretraining checkpoint, training with Hungarian matching using focal loss (weight=2.0, $\alpha$=0.25, $\gamma$=2.0), L1 loss (weight=5.0), and GIoU loss (weight=2.0). We use a per-GPU batch size of 4 with the default sampler.

**Text encoder and embedding generation.** Each image is represented by a {CLASS in MODALITY} prompt (see Figure 5), with categories deduplicated and sorted. We precompute embeddings with three frozen encoders (OpenCLIP ViT-B/32, BiomedCLIP, PubMedCLIP) by tokenizing and encoding the prompts in `eval()` mode on GPU with `torch.no_grad()`, saving one `.npy` per image. During finetuning, these embeddings are loaded per image and used to compute the contrastive alignment loss.

**Choice of Text.** We adopt a concise and unambiguous textual representation for each category. Specifically, we employ the {CLASS in MODALITY} template, which pairs the semantic name of the target class (CLASS) with its imaging domain (MODALITY). The complete set of 27 categories is provided in Figure 5. This formulation explicitly preserves both object semantics and modality context, without relying on additional descriptive phrases.

Table 8: Utilization of different text prompts. **Bold** = best per column.

| Text | AP | AP$_{50}$ | AP$_{75}$ | AP$_s$ | AP$_m$ | AP$_l$ |
|---|---|---|---|---|---|---|
| ✗ | 37.7 | 58.6 | 37.8 | 19.9 | 29.8 | 41.7 |
| {MODALITY} | 40.7 | 65.1 | 42.7 | 26.1 | 33.0 | 44.2 |
| {CLASS} | 41.0 | 65.4 | 43.2 | **27.6** | 33.6 | 44.5 |
| {CLASS in MODALITY} | **41.1** | **65.5** | **43.4** | 25.4 | **34.6** | **44.9** |

Table 9: Quantitative comparison of applying our method (MoCA + QueryREPA) on the DiffusionDet (Chen et al., 2023) model. **Bold** = best per column.

| Method | AP | AP$_{50}$ | AP$_{75}$ | AP$_s$ | AP$_m$ | AP$_l$ |
|---|---|---|---|---|---|---|
| DiffusionDet | 38.2 | 60.3 | 40.1 | 22.4 | 30.6 | 41.6 |
| DiffusionDet + Ours ($\mathcal{E}$: CLIP) | **40.2** | **65.4** | **41.6** | **26.0** | **34.5** | **42.6** |

## E    UTILIZATION OF DIFFERENT TEXT PROMPTS

In the proposed method, we choose a compact {CLASS in MODALITY} template for constructing text prompts (*e.g.*, "Nodule in lung CT", "Brain tumor in MRI"), so that each token explicitly encodes both the visual category and its imaging domain. To assess the suitability of this text design, we give an explicit ablation over prompt templates in Table 8.

Concretely, we compare three variants: {CLASS in MODALITY}, {CLASS} only, and {MODALITY} only. All three variants significantly improve AP over the detector baseline without any tokens. However, {CLASS in MODALITY} consistently achieves the best performance, {CLASS} is second-best, and {MODALITY} shows lowest performance of the three. This indicates that (i) the tokens indeed carry non-trivial semantic signal, since even very simple {CLASS} or {MODALITY} prompts provide measurable gains, and (ii) jointly encoding class and modality is empirically preferable to either component alone. In particular, the fact that {CLASS} > {MODALITY} suggests that the tokens capture fine-grained, class-specific distinctions beyond a coarse modality label, while the additional modality phrase further sharpens disentanglement. Thus, if the tokens were only encoding very rough modality semantics, then the {MODALITY}-only variant would be expected to perform similarly to {CLASS in MODALITY}, which is not what we observe.

## F    BEYOND DETR-STYLE MODELS

Our framework is highly general and can be applied whenever two mild conditions are met: (i) object-level embeddings are used, and (ii) these embeddings undergo self-attention. These requirements are very loose and are satisfied by most modern detectors, including CNN-style and diffusion-based methods.

For example, traditional CNN-style detectors (e.g., Faster R-CNN Ren et al. (2015), Mask R-CNN He et al. (2017)) originally relied on a large number of dense object candidates, which led to high training cost. More recent CNN-style detectors (e.g., Sparse R-CNN Sun et al. (2021), DiffusionDet Chen et al. (2023)) alleviate this inefficiency and improve performance by introducing a small set of object proposals together with attention-based iterative refinement. These object proposals are analogous to the object-level embeddings (*i.e.*, "object queries" in DETR-style methods) that we exploit in our work. Hence, our method can also be readily adapted for modern CNN-style detectors.

As a proof of concept, we show additional results applying our method (QueryREPA+MoCA) to the state-of-the-art detector DiffusionDet in Table 9. Note that DiffusionDet directly adopts the detection decoder from Sparse R-CNN and can therefore be viewed as a CNN-style detector. To integrate our method into DiffusionDet, we simply change its region proposals (akin to object proposals) into learnable embeddings and then apply our approach: we concatenate a modality token in the self-attention stage (MoCA) and perform QueryREPA pre-training. This lightweight adaptation substantially improves detection performance, for example boosting AP$_{50}$ from 60.3 to 65.4 (+5.1). These results demonstrate that our method can be easily incorporated into most modern detectors while still yielding significant performance gains.

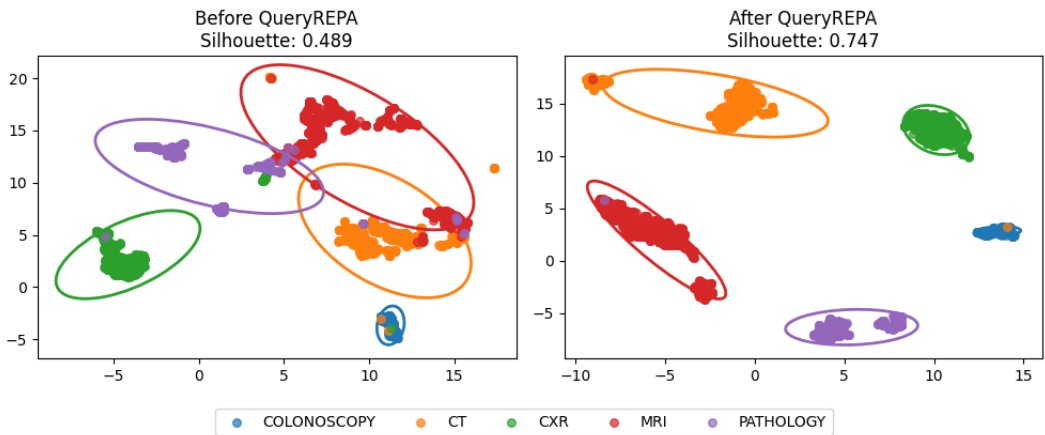

Figure 6: **QueryREPA incorporates modality-aware semantics into the query space.** UMAP embeddings of the query cluster mean $\bar{q}^{(l)}$ for various modalities are visualized. The visualizations show that the object queries before QueryREPA (left) are highly entangled, whereas object queries after QueryREPA (right) are well separated into disentangled clusters.

## G    ADDRESSING THE HETEROGENEITY OF REPRESENTATION SPACES

The "raw" query manifold is originally learned purely from heterogeneous image statistics. Our method is explicitly designed to address the heterogeneity of representation spaces across imaging modalities at the level of *object queries* in modern detectors. In our work, MoCA injects modality context into the query set by appending a text-derived modality token and mixing it with object queries. This converts the raw query manifold into a space where each query is conditioned on modality-aware semantic anchors. QueryREPA is further proposed as a way to reshape this manifold by aligning the mean query representation of each sample to its corresponding modality token, maximizing mutual information between queries and modality tokens. This effectively provides the detection decoder with more informative query initialization for downstream detection.

**Object Query Visualization Before/After QueryREPA**    QueryREPA guides the object queries toward a modality-aware semantic representation. As shown in Figure 6, the queries before QueryREPA are highly entangled and exhibit substantial overlap across modalities, reflecting the absence of modality-specific organization in the initial query space. After applying QueryREPA, the queries are appropriately mapped into disentangled regions. Such improvement is also evident quantitatively, with the silhouette score increasing from 0.489 to 0.747. These results indicate that QueryREPA reshapes the query space to incorporate modality-aware semantics, which in turn provides the decoder with more informative query initialization and facilitates more reliable reasoning across heterogeneous medical domains.

# H MORE QUALITATIVE RESULTS

We present additional qualitative results for each modality (CXR, MRI, Colonoscopy, CT, Pathology, and Pathology) in Figures 7–11. Each figure compares baseline detectors (Sparse R-CNN, GLIP, DiffusionDet, Grounding DINO, DINO) with our method, with blue boxes for baselines, green boxes for ground truth, and red boxes for our predictions.

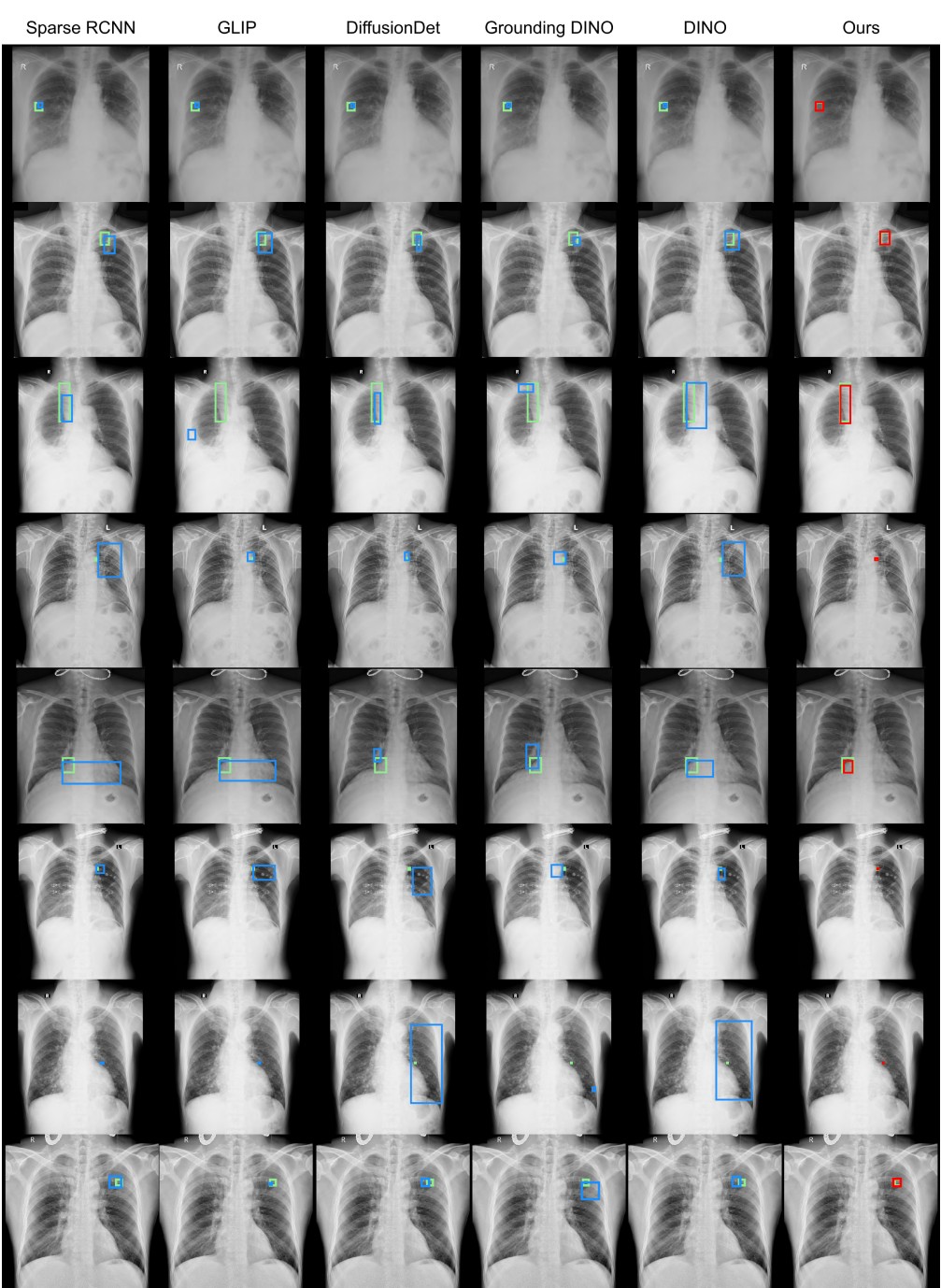

Figure 7: **Qualitative comparisons on chest X-ray images.** Blue boxes indicate predictions from baseline detectors (Sparse R-CNN, GLIP, DiffusionDet, Grounding DINO, DINO), Green boxes denote ground-truth annotations, and Red boxes show predictions from our method.

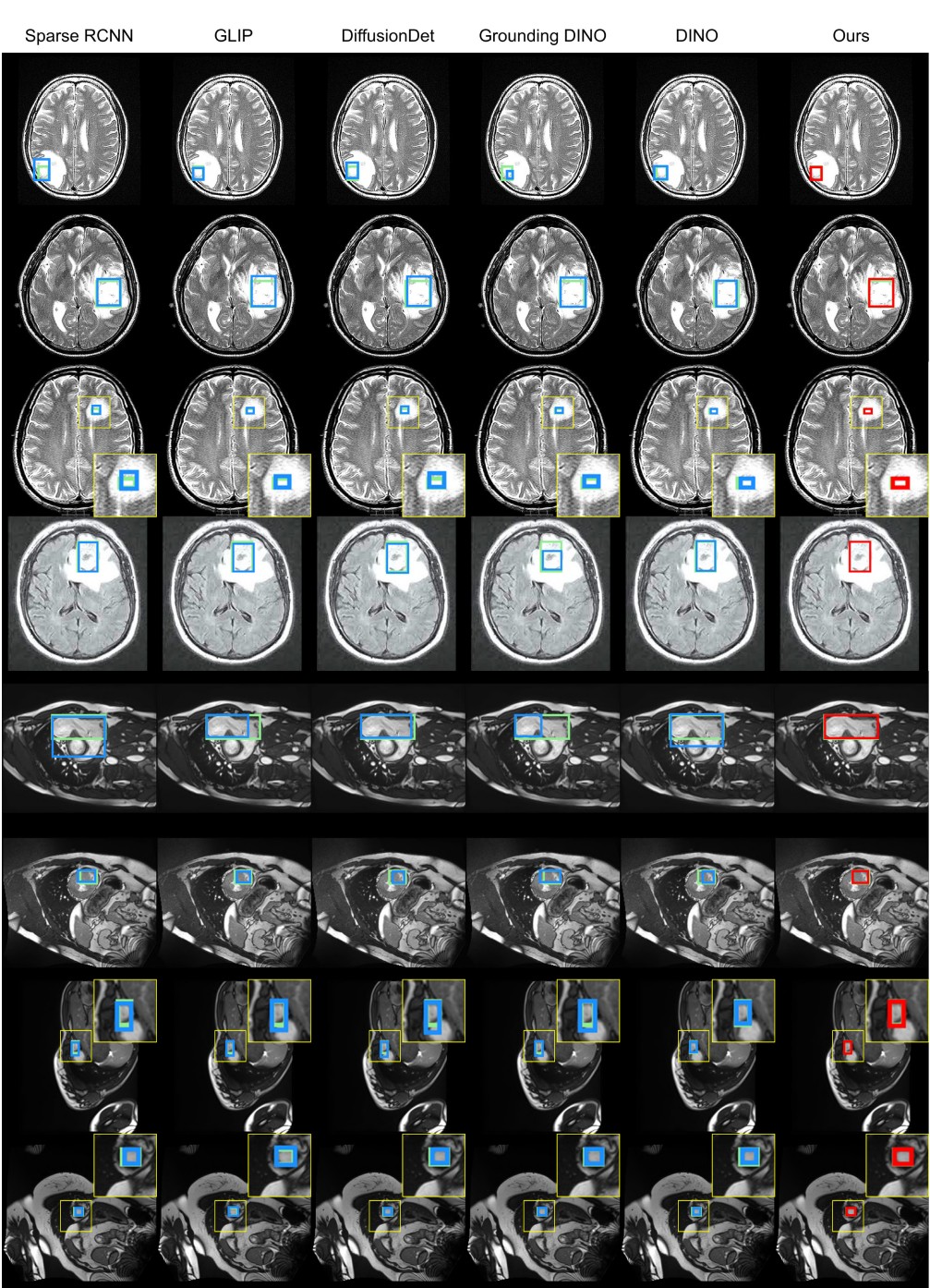

Figure 8: **Qualitative comparisons on MRI images.** Blue boxes indicate predictions from baseline detectors (Sparse R-CNN, GLIP, DiffusionDet, Grounding DINO, DINO), Green boxes denote ground-truth annotations, and Red boxes show predictions from our method.

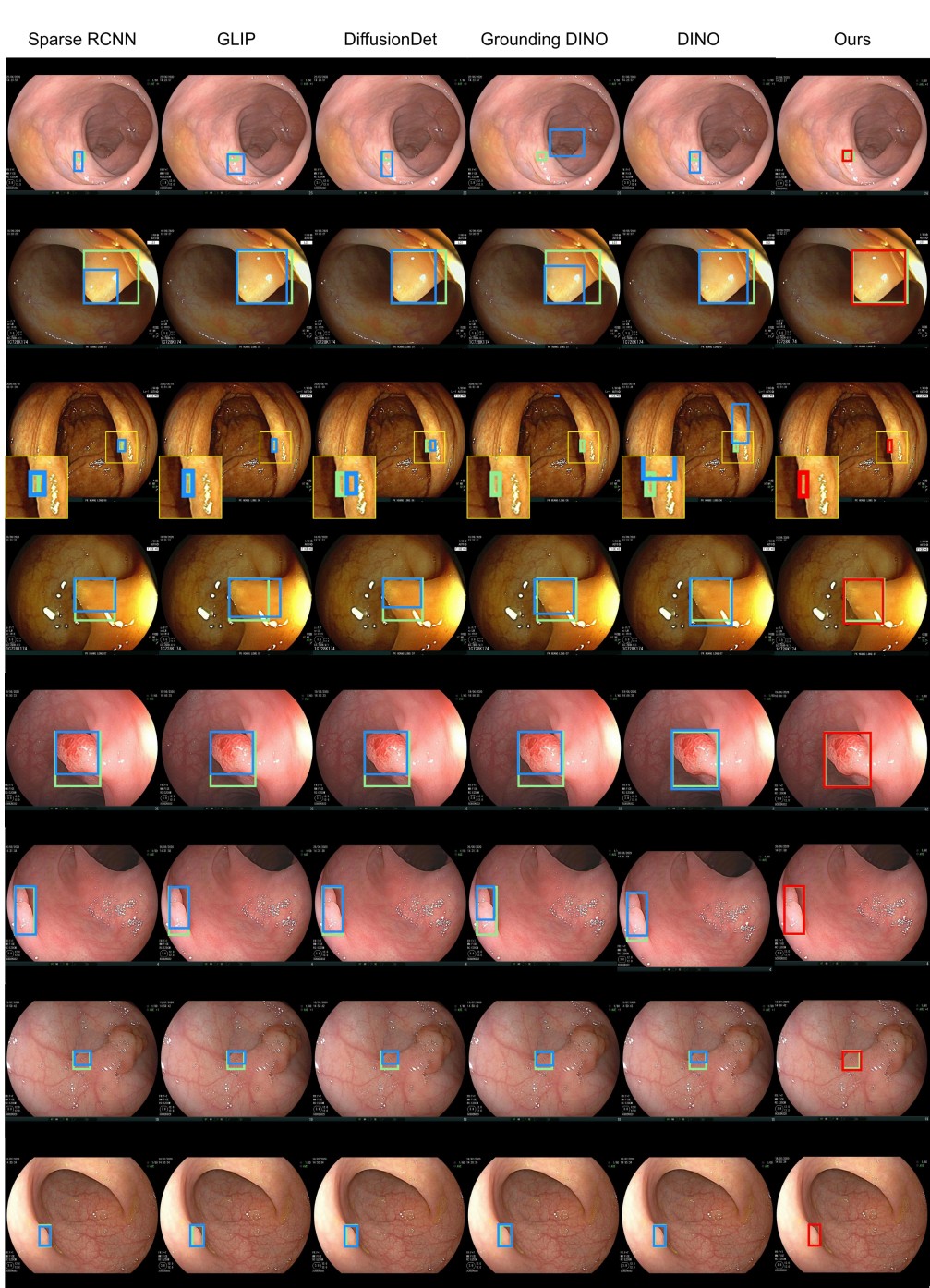

Figure 9: **Qualitative comparisons on Colonoscopy images.** Blue boxes indicate predictions from baseline detectors (Sparse R-CNN, GLIP, DiffusionDet, Grounding DINO, DINO), Green boxes denote ground-truth annotations, and Red boxes show predictions from our method.

Sparse RCNN    GLIP    DiffusionDet    Grounding DINO    DINO    Ours

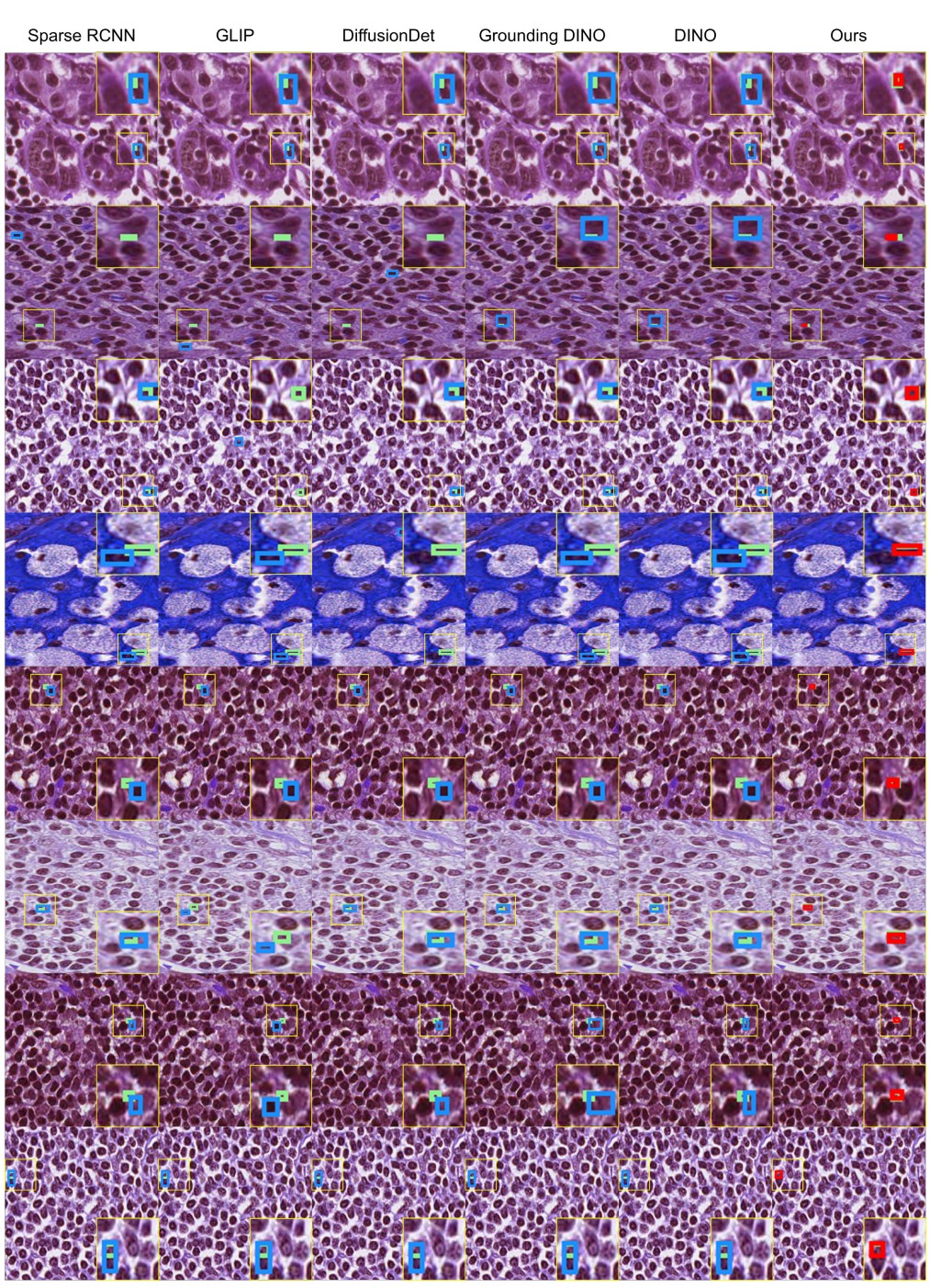

Figure 10: **Qualitative comparisons on Pathology images.** Blue boxes indicate predictions from baseline detectors (Sparse R-CNN, GLIP, DiffusionDet, Grounding DINO, DINO), Green boxes denote ground-truth annotations, and Red boxes show predictions from our method.

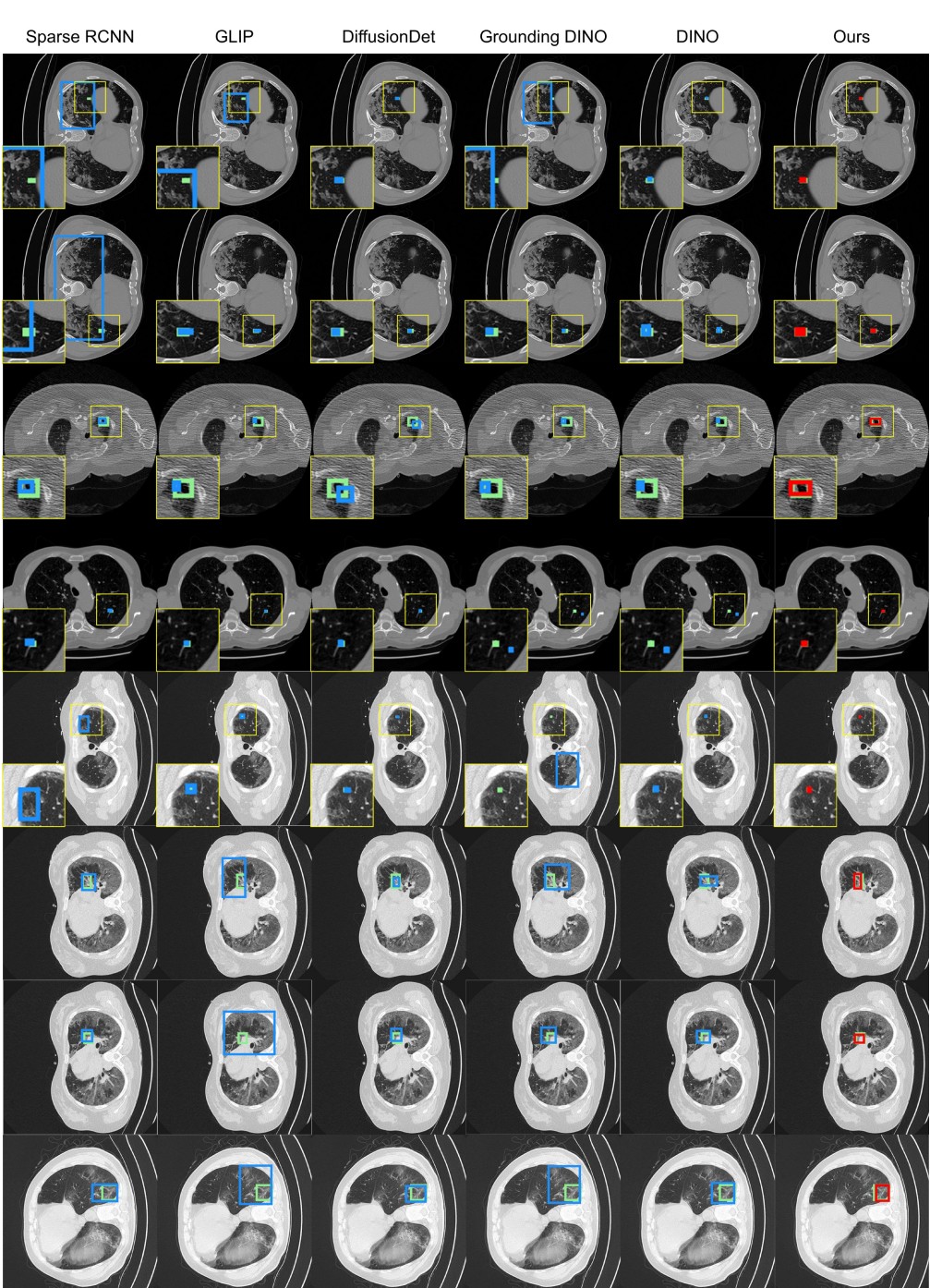

Figure 11: **Qualitative comparisons on CT images.** Blue boxes indicate predictions from baseline detectors (Sparse R-CNN, GLIP, DiffusionDet, Grounding DINO, DINO), Green boxes denote ground-truth annotations, and Red boxes show predictions from our method.

# I ALGORITHM FOR QUERYREPA

---

**Algorithm 1** Pseudocode of QueryREPA pretraining in PyTorch-like style.

---

```
"""
<Input>
images      : (B,H,W,3)
text_prompts: B text prompts of the form "{CLASS} in {MODALITY}"
text_encoder: CLIP / BiomedCLIP / PubMedCLIP

<Output>
loss_qrepa: query representation alignment loss

# Dt: dimension of text embedding from the text encoder
# D : dimension of projected query/text features used for alignment
# Nq: number of learnable object queries per image
"""

def queryrepa_loss(images, text_prompts):

    # 1) Encode text prompts into modality tokens m
    m = text_encoder(text_prompts)                    # (B, Dt)
    m_proj = proj_text(m)                             # (B, D)

    # 2) Forward backbone and decoder up to layer l
    feats = detector_backbone(images)
    learnable_object_queries = init_queries(B, Nq)    # (B, Nq, D)
    queries_list = detector_decoder(
        learnable_object_queries, m_proj, feats)
    q_l = queries_list[l]                             # queries at layer l

    # 3) Mean over all queries
    q_mean = q_l.mean(dim=1)                          # (B, D)

    # 4) Project queries into modality-token space
    q_proj = proj_query(q_mean)                       # (B, Dt)

    # 5) InfoNCE-style contrastive alignment between q_proj and m
    q_norm = F.normalize(q_proj, dim=-1)
    m_norm = F.normalize(m, dim=-1)
    logits  = torch.matmul(q_norm, m_norm.T) / tau
    targets = torch.arange(B)

    loss_qrepa = F.cross_entropy(logits, targets)

    return loss_qrepa
```

---

# J THE USE OF LARGE LANGUAGE MODELS (LLMS)

The use of LLMs in this work was strictly limited to minor language polishing and did not influence research ideation, methodological design, or experimental analysis. The authors take full responsibility for the accuracy and integrity of all scientific content presented in this paper.

