# OpenReview forum: "Align Your Query: Representation Alignment for Multimodality Medical Object Detection"
_ICLR.cc/2026/Conference — Submitted to ICLR 2026_

### Official Review · Reviewer_pyJE · 2025-10-30

**Soundness:** 3
**Presentation:** 3
**Contribution:** 3
**Rating:** 4
**Confidence:** 5

**Summary:**

This paper introduces a framework that aligns internal object-query representations with modality context, allowing one unified detector to perform well across diverse imaging types. The proposed framework makes the detector “aware” of each modality through lightweight modality tokens, text embeddings such as “tumor in MRI” or “aortic enlargement in CXR.” In a short pretraining stage called QueryREPA, the model aligns its internal object queries with these tokens using contrastive learning, producing modality-aware query representations. During detection, a Multimodality Context Attention (MoCA) mechanism appends the modality token to the query set so the model can attend to modality context while predicting bounding boxes. Tested on a large mixed-modality dataset built from seven public sources, the method consistently outperformed strong baselines like DINO and Deformable DETR, improving average precision by 3–4 points with almost no extra computation, and yielding notably sharper and more accurate lesion localization.

This paper shows that a simple, modular alignment strategy, combining lightweight text-based modality tokens, a short contrastive pretraining, and a self-attention fusion step, can make standard detectors far more robust in multimodality medical settings.

**Strengths:**

- The paper is clearly written, with a logical flow from motivation to method and experiments. The figures (especially the MoCA and QueryREPA schematic) effectively illustrate the pipeline.

- The challenge of building a single detector that generalizes across medical imaging modalities (CXR, CT, MRI, pathology) is practically relevant and timely for generalist medical AI.

- The approach achieves modest but consistent AP gains across all tested modalities and object scales, demonstrating robustness and reproducibility of the improvement.

- The paper includes thorough training details, dataset descriptions, and implementation notes (appendices), supporting easy replication.

**Weaknesses:**

The core ingredients, contrastive representation alignment, modality tokens, and self-attention fusion, are not fundamentally new. Similar strategies already exist, such as in Grounding DINO. The conceptual novelty is thin; it’s a tidy engineering blend rather than a new paradigm.

Although described as “detector-agnostic,” the method still assumes a query-based transformer architecture. Extending it to CNN-style or diffusion-based detectors would likely require non-trivial adaptation.

The proposed “representation alignment” operates only at a coarse level, matching query and modality-token embeddings without ensuring semantic disentanglement or true feature sharing across modalities. As a result, the model may still overfit modality-specific cues rather than learning transferable visual structure.

Although, the authors do not explicitly claim reasoning ability, the scope remains limited to detection only. The model learns context alignment, not semantic reasoning, it cannot interpret relationships between findings (e.g., “lesion near the diaphragm”) or answer higher-level questions, and therefore remains a low-level detector.

**Questions:**

1. The paper combines contrastive alignment, modality tokens, and self-attention fusion, all of which have been explored in prior works (e.g., Grounding DINO, REPA). Could the authors clarify what is fundamentally new in this combination beyond engineering integration?

2. The alignment between query and modality-token embeddings seems to operate at a coarse global level. Have you examined whether the model actually learns shared semantic subspaces across modalities (e.g., via visualization or clustering metrics)?

3. Your framework focuses on aligning modality-aware object queries for improved detection across heterogeneous medical imaging types. Given the recent progress of unified multimodal models such as Qwen-VL and GPT-4V, which can already perform vision–language reasoning and coarse localization, how would your approach compare to adapting such a large multimodal foundation model for structured object detection (e.g., by adding a query-alignment or visual token–to–query head)? In your view, what are the trade-offs between starting from a domain-specific alignment framework like MoCA + QueryREPA versus extending a general VLM into a precise medical detector in terms of localization accuracy, data efficiency, and scalability?

---

> ### Author Response · Authors · 2025-11-25
>
> **W1, Q1. The paper combines contrastive alignment, modality tokens, and self-attention fusion, all of which have been explored in prior works (e.g., Grounding DINO, REPA). Could the authors clarify what is fundamentally new in this combination beyond engineering integration?**
>
> **A.** We have updated the **related works section** of our manuscript to give a more thorough explanation about existing representation alignment methods and explain their core differences to our method. To clarify, existing representation alignment methods for object detection can be roughly grouped into three paradigms: **(i) backbone-level alignment** of global image features, where contrastive or CLIP-style objectives act only on outputs of the *image encoder* and do not modify the logic of the detection model at all (e.g., MoCo [1], SimCLR [2], SwAV [3], ConVIRT [4]); **(ii) region-centric alignment**, which aligns *local* pre-defined region or proposal features to text descriptions (*e.g.*, RegionCLIP [5], DITO [6]); and **(iii) query-level alignment (ours)**, which directly aligns object-level embeddings or “queries” to compact modality tokens inside the decoder. Therefore, our framework differs from existing works in both what is being aligned *and* the target of the alignment. Thus, existing representation alignment methods are largely *complementary* to our method: in principle, it would be possible to pretrain the image encoder with backbone-level alignment and afterwards apply our query-level alignment.
>
> Furthermore, we demonstrate the potential of QueryREPA by showing how pretraining on extra images without bounding box labels can increase the final detection performance, in **Section 4.3** and **Table 4** of our updated manuscript. In the experiments, we observe that **including images without bounding-boxes during QueryREPA pretraining provides measurable performance gains**. Thus, QueryREPA *mitigates annotation scarcity* by turning otherwise underutilized, weakly labeled medical data into a beneficial resource that improves detection performance. This provides an entirely new research avenue for performing weakly supervised or unsupervised learning in detection models. Whereas existing unsupervised learning methods have largely focused on *image features*, our approach instead opens an intriguing path toward unsupervised learning directly on *object queries*.
>
> **References**
>
> [1] MoCo: Momentum Contrast for Unsupervised Visual Representation Learning, CVPR 2020
>
> [2] SimCLR: A Simple Framework for Contrastive Learning of Visual Representations, ICML 2020
>
> [3] Unsupervised Learning of Visual Features by Contrasting Cluster Assignments, NeurIPS 2020
>
> [4] ConVIRT: Contrastive Learning of Medical Visual Representations from Paired Images and Text, Nature Biomedical Engineering 2021
>
> [5] RegionCLIP: Region-Based Language–Image Pretraining, CVPR 2023
>
> [6] Region-centric Image-Language Pretraining for Open-Vocabulary Detection, ECCV 2024

---

> > ### Author Response · Authors · 2025-11-25
> >
> > **W2. Although described as “detector-agnostic,” the method still assumes a query-based transformer architecture. Extending it to CNN-style or diffusion-based detectors would likely require non-trivial adaptation.**
> >
> > **A.** We clarify that our framework is highly general and can be applied whenever two mild conditions are met: (i) object-level embeddings are used, and (ii) these embeddings undergo self-attention. These requirements are very loose and are satisfied by most modern detectors, *including CNN-style and diffusion-based methods*.
> >
> > For example, traditional CNN-style detectors (e.g., Faster R-CNN [1], Mask R-CNN [2]) originally relied on a large number of dense object candidates, which led to high training cost. More recent CNN-style detectors (e.g., Sparse R-CNN [3], DiffusionDet [4]) alleviate this inefficiency and improve performance by introducing *a small set of object proposals* together with *attention-based iterative refinement*. These object proposals are analogous to the object-level embeddings (i.e., “object queries” in DETR-style methods) that we exploit in our work. Hence, our method can also be readily adapted for modern CNN-style detectors.
> >
> > As a proof of concept, we show additional results applying our method (QueryREPA+MoCA) to the state-of-the-art detector DiffusionDet. Note that DiffusionDet directly adopts the detection decoder from Sparse R-CNN and can therefore be viewed as a CNN-style detector. To integrate our method into DiffusionDet, we simply change its region proposals (akin to object proposals) into learnable embeddings and then apply our approach: we concatenate a modality token in the self-attention stage (MoCA) and perform QueryREPA pre-training. The results are summarized below:
> >
> > | Method                         |   AP  | AP₅₀ | AP₇₅ | APₛ  | APₘ  | APₗ  |
> > |-------------------------------|:------:|:------:|:------:|:------:|:------:|:------:|
> > | DiffusionDet                  | 38.2 | 60.3 | 40.1 | 22.4 | 30.6 | 41.6 |
> > | DiffusionDet + Ours (*E*: CLIP) | **40.2** | **65.4** | **41.6** | **26.0** | **34.5** | **42.6** |
> >
> > This lightweight adaptation *substantially improves detection performance*, for example boosting AP₅₀ from 60.3 to 65.4 (+5.1). These results demonstrate that our method can be easily incorporated into most modern detectors while still yielding significant performance gains. We have included these results in **Appendix F** of our updated manuscript.
> >
> > **References**
> >
> > [1] Faster R-CNN: Towards Real-Time Object Detection with Region Proposal Networks, NeurIPS 2015
> >
> > [2] Mask R-CNN, ICCV 2017
> >
> > [3] Sparse R-CNN: End-to-end object detection with learnable proposals, CVPR 2021
> >
> > [4] DiffusionDet: Diffusion Model for Object Detection, ICCV 2023
> >
> > ---
> >
> > **W3, Q2. The alignment between query and modality-token embeddings seems to operate at a coarse global level. Have you examined whether the model actually learns shared semantic subspaces across modalities (e.g., via visualization or clustering metrics)?**
> >
> > **A.** Our method is explicitly designed to address the heterogeneity and disjointness of representation spaces across imaging modalities at the level of object queries in modern detectors. In our updated manuscript, we clarify this and provide a visualization of *object queries* before/after QueryREPA in **Appendix G** and **Figure 6**.
> >
> > The “raw” query manifold is originally learned purely from heterogeneous image statistics. In our work, **MoCA** injects modality context into the query set by appending a text-derived modality token and mixing it with object queries. This converts the raw query manifold into a space where each query is conditioned on modality-aware semantic anchors. **QueryREPA** is further proposed as a way to reshape this manifold by aligning the mean query representation of each sample to its corresponding modality token, maximizing mutual information between queries and modality tokens.
> >
> > We further visualize the query representations *before* and *after* QueryREPA using UMAP in **Figure 6** of our updated manuscript. The pre-QueryREPA queries are highly entangled and exhibit substantial overlap across modalities, indicating that the initial query space does not reflect the heterogeneous modalities. After applying QueryREPA, the query representations are mapped into well-separated, modality-aware clusters. Furthermore, the silhouette score of the query clusters increases from 0.489 to 0.747. These results demonstrate that our method indeed reduces the “disjoint representation spaces” issue by organizing query representations into a structured, modality-aware space that is shared across the heterogeneous multimodality dataset.

---

> > > ### Author Response · Authors · 2025-11-25
> > >
> > > **W4. Although the authors do not explicitly claim reasoning ability, the scope remains limited to detection only. The model cannot interpret relationships between findings or answer higher-level questions, and therefore remains a low-level detector.**
> > >
> > > **A.** Our goal in this work is indeed *multimodality medical object detection* (i.e., we aim to improve the robustness and accuracy of detectors when trained on mixed multimodality datasets), rather than developing a full semantic reasoning or report-generation system. Our contributions (modality tokens, MoCA, and QueryREPA) are explicitly targeted at aligning and enriching *object-query representations* within the detector.
> > >
> > > We agree that our model does not explicitly represent relational statements such as “lesion near the diaphragm” or answer high-level clinical questions, and we did not intend such capabilities. In this work, our focus is on **fast and robust multimodality object detection**: the design of a *single* shared text token and lightweight QueryREPA is intentional, so that we can enrich object queries with modality-aware context **without incurring the large computational overhead** that would come from heavier reasoning modules or report-generation heads (we show the minimal increase for decoder head latency in **Table 3**). We view our method as a strong, efficient perception front-end that can be plugged into downstream systems for relational reasoning, report generation, or question answering. Extending our framework with explicit modeling of relationships between findings and higher-level clinical reasoning is an interesting and important direction, but is orthogonal to our current scope and we leave it for future work.
> > >
> > > ---
> > >
> > > **Q3. How would your approach compare to adapting such a large multimodal foundation model for structured object detection? In your view, what are the trade-offs between starting from a domain-specific alignment framework like MoCA + QueryREPA versus extending a general VLM into a precise medical detector in terms of localization accuracy, data efficiency, and scalability?**
> > >
> > > **A.** We appreciate this insightful question and agree that adapting large multimodal foundation models such as Qwen-VL or GPT-4V for medical detection is an exciting direction. Our work, however, targets a different and complementary design point: accurate, data-efficient, and computationally lightweight detection across heterogeneous medical imaging modalities.
> > >
> > > First, current multimodal foundation models are pre-trained predominantly on natural images and generic web text, and their visual tokenization and attention mechanisms are optimized for coarse localization and open-ended reasoning, not for *exhaustive, small-scale, multi-lesion detection* in domain-specific modalities such as CXR, CT, MRI, endoscopy, and pathology. Adapting them into precise medical detectors would still require a comparable amount of box-level supervision and careful fine-tuning, and it is unclear whether their generic visual priors transfer optimally to subtle, modality-specific patterns (e.g., faint nodules or small polyps). In contrast, our framework assumes a strong, task-specific detection backbone and focuses on **modality-aware query alignment (MoCA + QueryREPA)** to address the heterogeneous-modality setting, which we show leads to consistent gains in detection accuracy.
> > >
> > > Second, multimodal foundation models are orders of magnitude larger and more computationally expensive than the detectors considered in this work. Using them as detection backbones would significantly increase memory and latency, which is problematic for typical workflows where thousands of images must be processed daily under strict latency and resource constraints. Our design deliberately keeps the alignment machinery lightweight so that we can improve multimodality detection with **minimal overhead** relative to the base detector, making integration into existing clinical pipelines more realistic.
> > >
> > > Overall, we view the reviewer’s question as pointing to an important and complementary research direction rather than a competing paradigm. Large multimodal models excel at broad vision-language reasoning, while our approach is explicitly optimized for precise detection. Nonetheless, exploring the utilization of multimodal foundation models would be an interesting and promising direction for future research.

---

### Official Review · Reviewer_MW9v · 2025-10-31

**Soundness:** 2
**Presentation:** 2
**Contribution:** 2
**Rating:** 4
**Confidence:** 5

**Summary:**

The paper proposes Align Your Query, a detector-agnostic framework for multimodality medical object detection that introduces Multimodality Context Attention (MoCA) to fuse text-derived modality tokens with object queries and QueryREPA contrastive pretraining to align query representations.

**Strengths:**

1.The paper is clearly written and well organized, making the technical background, proposed methods, and experimental results easy to follow.

2.The idea of aligning DETR-style object queries with text-derived modality tokens is conceptually interesting and demonstrates moderate empirical gains in multimodality medical detection.

**Weaknesses:**

1.The technical novelty is limited, as the work largely adheres to existing DETR-style frameworks and primarily adds lightweight extensions rather than proposing a fundamentally new detection paradigm.

2.The proposed QueryREPA pretraining introduces additional complexity without clear evidence that it significantly improves beyond standard multimodal contrastive alignment methods.

3.The evaluation lacks sufficient analysis on large-scale or real clinical datasets, limiting confidence in the method’s generalizability and real-world impact.

4.The paper’s ablation and visualization results, while supportive, do not thoroughly isolate the individual contributions of MoCA and QueryREPA, making the source of performance gains somewhat ambiguous.

**Questions:**

1.How does the proposed method go beyond prior DETR-style architectures to offer genuine methodological novelty?

2.Can the authors provide clearer evidence that QueryREPA offers distinct benefits over standard contrastive pretraining approaches?

3.Have the authors considered evaluating on larger or clinically diverse datasets to better demonstrate real-world generalization?

4.Can the authors expand the ablation studies to more clearly separate the effects of MoCA and QueryREPA?

**Details Of Ethics Concerns:**

As mentioned in the paper: We adhere to the ICLR Code of Ethics. Our study uses only publicly available medical imaging
datasets (VinBigData, COVID-19 CT, LIDC-IDRI, NeoPolyp, BR35H, ACDC, MoNuSeg) contain-
ing no personally identifiable information. No new human-subject data were collected, and thus
no IRB approval was required.

---

> ### Author Response · Authors · 2025-11-25
>
> **W1, Q1. The technical novelty is limited, as the work largely adheres to existing DETR-style frameworks and primarily adds lightweight extensions rather than proposing a fundamentally new detection paradigm.**
>
> **A.** Our framework differs from existing works in both *what is being aligned* and *what is the target of the alignment*. We have updated the **related works section** of our manuscript to give a more thorough explanation about existing representation alignment methods and explain their core differences to our method. To clarify, existing representation alignment methods for object detection can be roughly grouped into three paradigms: **(i) backbone-level alignment** of global image features, where contrastive or CLIP-style objectives act only on outputs of the *image encoder* and do not modify the logic of the detection model at all (e.g., MoCo [1], SimCLR [2], SwAV [3], ConVIRT [4]); **(ii) region-centric alignment**, which aligns *local* pre-defined region or proposal features to text descriptions (*e.g.*, RegionCLIP [5], DITO [6]); and **(iii) query-level alignment (ours)**, which directly aligns object-level embeddings or “queries” to compact modality tokens inside the decoder. Therefore, existing representation alignment methods are largely *complementary* to our method: in principle, it would be possible to pretrain the image encoder with backbone-level alignment and afterwards apply our query-level alignment.
>
> Furthermore, we clarify that our framework is highly general and can be applied whenever two mild conditions are met: (i) object-level embeddings are used, and (ii) these embeddings undergo self-attention. These requirements are very loose and are satisfied by most modern detectors, *not only DETR-style frameworks*. As a proof of concept, we show additional results applying our method (QueryREPA+MoCA) to the state-of-the-art detector DiffusionDet [7]. Note that DiffusionDet directly adopts the detection decoder from Sparse R-CNN [8] and can therefore be viewed as a CNN-style detector. To integrate our method into DiffusionDet, we simply change its region proposals (akin to object proposals) into learnable embeddings and then apply our approach: we concatenate a modality token in the self-attention stage (MoCA) and perform QueryREPA pre-training. The results are summarized below:
>
> | Method                         |   AP  | AP₅₀ | AP₇₅ | APₛ  | APₘ  | APₗ  |
> |-------------------------------|:------:|:------:|:------:|:------:|:------:|:------:|
> | DiffusionDet                  | 38.2 | 60.3 | 40.1 | 22.4 | 30.6 | 41.6 |
> | DiffusionDet + Ours | **40.2** | **65.4** | **41.6** | **26.0** | **34.5** | **42.6** |
>
> This lightweight adaptation *substantially improves detection performance*, for example boosting AP₅₀ from 60.3 to 65.4 (+5.1). These results demonstrate that our method can be easily incorporated into most modern detectors while still yielding significant performance gains. We have included these results in **Appendix F** of our updated manuscript.
>
> **References**
>
> [1] MoCo: Momentum Contrast for Unsupervised Visual Representation Learning, CVPR 2020
>
> [2] SimCLR: A Simple Framework for Contrastive Learning of Visual Representations, ICML 2020
>
> [3] Unsupervised Learning of Visual Features by Contrasting Cluster Assignments, NeurIPS 2020
>
> [4] ConVIRT: Contrastive Learning of Medical Visual Representations from Paired Images and Text, Nature Biomedical Engineering 2021
>
> [5] RegionCLIP: Region-Based Language–Image Pretraining, CVPR 2023
>
> [6] Region-centric Image-Language Pretraining for Open-Vocabulary Detection, ECCV 2024
>
> [7] DiffusionDet: Diffusion Model for Object Detection, ICCV 2023
>
> [8] Sparse R-CNN: End-to-end object detection with learnable proposals, CVPR 2021

---

> > ### Author Response · Authors · 2025-11-25
> >
> > **W2, Q2. The proposed QueryREPA pretraining introduces additional complexity without clear evidence that it significantly improves beyond standard multimodal contrastive alignment methods.**
> >
> > **A.** As explained in the answer above, existing representation alignment methods are largely *complementary* to our method. For example, it is possible to contrastively align the image encoder backbone, and then apply our query-level alignment afterwards. Thus, QueryREPA provides an *new research avenue* for performing contrastive alignment on object queries.
> >
> > Furthermore, while QueryREPA introduces additional complexity before end-to-end detection training, this step offers a practical advantage in real-world medical settings where bounding box annotations are scarce. QueryREPA performs *representation alignment at the query level*, requiring only image-level class labels and **no bounding-box supervision during the pretraining stage**. This makes it particularly suitable for large medical datasets in which detailed annotations are limited.
> >
> > To demonstrate the potential of QueryREPA, we show below how pretraining on extra images *without* bounding box labels can increase the final detection performance. As an example, we additionally leverage the widely used **NIH Chest X-ray** dataset, which contains over 112,000 images but **provides bounding-box annotations for fewer than 1,000 cases**. This leaves the vast majority of images unusable for conventional detection pretraining. In the table below, we compare between different datasets used for QueryREPA, while keeping the datasets used for detection training the same.
> >
> > | QueryREPA Dataset | Detection Training Dataset | Total AP | Total AP₅₀ | CXR AP | CXR AP₅₀ | MRI AP | MRI AP₅₀ | Colonoscopy AP | Colonoscopy AP₅₀ | Pathology AP | Pathology AP₅₀ | CT AP | CT AP₅₀ |
> > |:------------------:|:----------------------------:|:---------:|:-----------:|:-------:|:---------:|:-------:|:---------:|:----------------:|:-----------------:|:-------------:|:---------------:|:------:|:--------:|
> > | ✗                | (2) + (3)                  | 38.5     | 62.0       | 19.1   | 41.5     | 77.4   | 96.8     | 68.7            | 86.8             | 53.4         | 83.0           | 34.2  | 66.2    |
> > | (2) + (3)        | (2) + (3)                  | 39.1     | 62.9       | 19.2   | 42.5     | 77.6   | 96.6     | 70.4            | 85.7             | **55.7**     | **85.9**       | **34.6** | 67.7  |
> > | (1) + (2) + (3)  | (2) + (3)                  | **39.4** | **63.3**   | **19.6** | **42.6** | **78.0** | **97.1** | **72.6**        | **89.9**         | 54.7         | 84.8           | 34.1  | **68.0** |
> >
> >
> > Specifically, the datasets are: (1) NIH with only class labels (2) NIH with bounding box labels (3) VinBigData with bounding box labels. These results are also included in **Table 4** of our updated manuscript.
> >
> > In the experiment above, we observe that **including images without bounding-boxes during QueryREPA pretraining provides measurable performance gains**. Thus, although QueryREPA adds an explicit pretraining step, it *mitigates annotation scarcity* by turning otherwise underutilized, weakly labeled medical data into a beneficial resource that improves detection performance. This provides an entirely new research avenue for performing weakly supervised or unsupervised learning in detection models. Whereas existing unsupervised learning methods have largely focused on *image features*, our approach instead opens an intriguing path toward unsupervised learning directly on *object queries*.

---

> > > ### Author Response · Authors · 2025-11-25
> > >
> > > **W3, Q3. The evaluation lacks sufficient analysis on large-scale or real clinical datasets, limiting confidence in the method’s generalizability and real-world impact.**
> > >
> > > **A.** Our demonstration on the **NIH Chest X-ray** dataset for the answer above (also included in **Table 4** of our updated manuscript) gives insight into the generalizability and real-world impact of our method. To reiterate, the **NIH Chest X-ray** dataset is a **large-scale, real clinical dataset** of over 112,000 images from over 30,000 unique patients. However, bounding-box annotations are existent for fewer than 1,000 cases, leaving the vast majority of images unusable for conventional detection pretraining. In the table above, we demonstrate how QueryREPA effectively improves detection performance in this scenario, and actually makes possible the usage of numerous unlabeled data. Thus, our proposed method shows both high generalizability and high scalability.
> > >
> > > ---
> > >
> > > **W4, Q4. The paper’s ablation and visualization results, while supportive, do not thoroughly isolate the individual contributions of MoCA and QueryREPA.**
> > >
> > > **A.** We provide a thorough ablation of the individual contributions in **Section 4.4** and **Table 5** of our updated manuscript. Ablation results are summarized below:
> > >
> > > | QueryREPA | MoCA |   AP  | AP₅₀ | AP₇₅ | APₛ  | APₘ  | APₗ  |
> > > |:---------:|:----:|------:|------:|------:|------:|------:|------:|
> > > | ✗         | ✗    | 37.7 | 58.6 | 37.8 | 19.9 | 29.8 | 41.7 |
> > > | ✓         | ✗    | 38.1 | 58.4 | 40.9 | 21.0 | 29.8 | 42.3 |
> > > | ✗         | ✓    | 40.8 | 65.1 | **43.4** | 25.1 | 33.5 | **45.2** |
> > > | ✓         | ✓    | **41.1** | **65.5** | **43.4** | **25.4** | **34.6** | 44.9 |
> > >
> > > These experiments confirm that both MoCA and QueryREPA independently contribute to the performance improvements, and that their combination yields the best results due to their synergistic effect.
> > > Furthermore, **Table 4** (i.e., the table for QueryREPA on NIH Chest X-ray data) in our updated manuscript further validates the effectiveness of QueryREPA by varying the datasets used for QueryREPA pretraining while keeping the datasets used for detection training fixed.

---

> > > > ### Comment · Reviewer_MW9v · 2025-11-27
> > > >
> > > > I appreciate the authors’ rebuttal and would also like to see further discussion from the other reviewers. For now, after carefully reading the rebuttal and the other reviews, I will maintain my original score of 4.

---

> > > > > ### Author Response · Authors · 2025-11-27
> > > > >
> > > > > We sincerely thank the reviewer for the careful consideration of our rebuttal and for the continued interest in the discussion. We have further improved the clarity of QueryREPA by revising Figure 1, refining the description in Section 3.3, and adding an explicit algorithmic pseudocode in Appendix I. We respectfully invite the reviewer to refer to these updates in the revised manuscript.

---

### Official Review · Reviewer_osNV · 2025-10-31

**Soundness:** 2
**Presentation:** 3
**Contribution:** 2
**Rating:** 4
**Confidence:** 3

**Summary:**

The paper proposes a DETR-based object detection framework tailored for multimodal medical imaging, introducing two key components, MoCA  and Query REPA, that reportedly enhance the performance of existing detection models in medical scenarios.

**Strengths:**

1. The proposed method requires no modification to the underlying detector architecture, incurs negligible computational overhead, and is compatible with other approaches.
2. Experiments span multiple medical imaging modalities (e.g., X-ray, MRI, ultrasound), demonstrating the model’s generalizability. Both quantitative results and qualitative visualizations are provided, offering clear and intuitive evidence of effectiveness.

**Weaknesses:**

### **Major Weaknesses**

1. The paper mentions “heterogeneous statistics and disjoint representation spaces” at the beginning of the abstract, but the main text does not further elaborate on whether the proposed method addresses this issue, nor does it explain how it does so.
2. Regarding the paper’s two core contributions, MoCA and QueryREPA, although the authors describe them in a very elaborate manner, I do not perceive them as particularly novel. But my familiarity with DETR-based methods is limited, so I would like to hear the opinions of other reviewers on this matter.
3. Proposition 1 appears rather trivial. It is widely recognized that increasing the batch size can enhance the effectiveness of the InfoNCE loss.

### Minor Weaknesses

1. The baselines are somewhat outdated. The paper should include more recent and competitive baselines.
2. It seems that each modality uses a separate, dedicated image encoder. The authors should provide clearer clarification on this point.
3. Suggest to cite the following paper:

    Liu, Y., Xi, S., Liu, S., Ding, H., Jin, C., Zhong, C., ... & Shen, Y. (2025). Multimodal Medical Image Binding via Shared Text Embeddings. *arXiv preprint arXiv:2506.18072*.

**Questions:**

The authors need to address the issues I raised in the Weaknesses section, primarily clarifying why the proposed method works and articulating the core novelty of the approach.

I will base my final scoring decision on the authors’ responses to these points, as well as feedback from other reviewers.

---

> ### Author Response · Authors · 2025-11-25
>
> **W1. The paper mentions “heterogeneous statistics and disjoint representation spaces” at the beginning of the abstract, but the main text does not further elaborate on whether the proposed method addresses this issue, nor does it explain how it does so.**
>
> **A**. We thank the reviewer for pointing out this missing connection. Our method is explicitly designed to address the heterogeneity and disjointness of representation spaces across imaging modalities at the level of *object queries* in modern detectors. In our updated manuscript, we clarify this and provide a visualization of object queries before/after QueryREPA in **Appendix G** and **Figure 6**.
>
> The “raw” query manifold is originally learned purely from heterogeneous image statistics. In our work, **MoCA** injects modality context into the query set by appending a text-derived modality token and mixing it with object queries. This converts the raw query manifold into a space where each query is conditioned on modality-aware semantic anchors. **QueryREPA** is further proposed as a way to reshape this manifold by aligning the mean query representation of each sample to its corresponding modality token, maximizing mutual information between queries and modality tokens.
>
> We further visualize the query representations *before* and *after* QueryREPA using UMAP in **Figure 6** of our updated manuscript. The pre-QueryREPA queries are highly entangled and exhibit substantial overlap across modalities, indicating that the initial query space does not reflect the heterogeneous modalities. After applying QueryREPA, the query representations are mapped into well-separated, modality-aware clusters. Furthermore, the silhouette score of the query clusters increases from 0.489 to 0.747. These results demonstrate that our method indeed reduces the “disjoint representation spaces” issue by organizing query representations into a structured, modality-aware space that is shared across the heterogeneous multimodality dataset.

---

> > ### Author Response · Authors · 2025-11-25
> >
> > **W2. Regarding the paper’s two core contributions, MoCA and QueryREPA, although the authors describe them in a very elaborate manner, I do not perceive them as particularly novel.**
> >
> > **A.** Our framework differs from existing works in both *what is being aligned* and *what is the target of the alignment*. We have updated the **related works section** of our manuscript to give a more thorough explanation about existing representation alignment methods and explain their core differences to our method. To clarify, existing representation alignment methods for object detection can be roughly grouped into three paradigms: **(i) backbone-level alignment** of global image features, where contrastive or CLIP-style objectives act only on outputs of the *image encoder* and do not modify the logic of the detection model at all (e.g., MoCo [1], SimCLR [2], SwAV [3], ConVIRT [4]); **(ii) region-centric alignment**, which aligns *local* pre-defined region or proposal features to text descriptions (e.g., RegionCLIP [5], DITO [6]); and **(iii) query-level alignment (ours)**, which directly aligns object-level embeddings or “queries” to compact modality tokens inside the decoder. Therefore, existing representation alignment methods are largely *complementary* to our method: in principle, it would be possible to pretrain the image encoder with backbone-level alignment and afterwards apply our query-level alignment.
> >
> > Furthermore, QueryREPA is a novel technique that requires only image-level class labels and **no bounding-box supervision during the pretraining stage**. This offers a practical advantage in real-world medical settings where bounding box annotations are scarce. To demonstrate the potential of QueryREPA, we show below how pretraining on extra images *without* bounding box labels can increase the final detection performance. As an example, we additionally leverage the widely used **NIH Chest X-ray** dataset, which contains over 112,000 images but **provides bounding-box annotations for fewer than 1,000 cases**. This leaves the vast majority of images unusable for conventional detection pretraining. In the table below, we compare between different datasets used for QueryREPA, while keeping the datasets used for detection training the same.
> >
> > | QueryREPA Dataset | Detection Training Dataset | Total AP | Total AP₅₀ | CXR AP | CXR AP₅₀ | MRI AP | MRI AP₅₀ | Colonoscopy AP | Colonoscopy AP₅₀ | Pathology AP | Pathology AP₅₀ | CT AP | CT AP₅₀ |
> > |:------------------:|:----------------------------:|:---------:|:-----------:|:-------:|:---------:|:-------:|:---------:|:----------------:|:-----------------:|:-------------:|:---------------:|:------:|:--------:|
> > | ✗                | (2) + (3)                  | 38.5     | 62.0       | 19.1   | 41.5     | 77.4   | 96.8     | 68.7            | 86.8             | 53.4         | 83.0           | 34.2  | 66.2    |
> > | (2) + (3)        | (2) + (3)                  | 39.1     | 62.9       | 19.2   | 42.5     | 77.6   | 96.6     | 70.4            | 85.7             | **55.7**     | **85.9**       | **34.6** | 67.7  |
> > | (1) + (2) + (3)  | (2) + (3)                  | **39.4** | **63.3**   | **19.6** | **42.6** | **78.0** | **97.1** | **72.6**        | **89.9**         | 54.7         | 84.8           | 34.1  | **68.0** |
> >
> >
> > Specifically, the datasets are: (1) NIH with only class labels (2) NIH with bounding box labels (3) VinBigData with bounding box labels. These results are also included in **Table 4** of our updated manuscript.
> >
> > In the experiment above, we observe that **including images without bounding-boxes during QueryREPA pretraining provides measurable performance gains.** Thus, QueryREPA is a novel technique that *mitigates annotation scarcity* by turning otherwise underutilized, weakly labeled medical data into a beneficial resource that improves detection performance. This provides an entirely new research avenue for performing weakly supervised or unsupervised learning in detection models. Whereas existing unsupervised learning methods have largely focused on *image features*, our approach instead opens an intriguing path toward unsupervised learning directly on *object queries*.
> >
> > **References**
> >
> > [1] MoCo: Momentum Contrast for Unsupervised Visual Representation Learning, CVPR 2020
> >
> > [2] SimCLR: A Simple Framework for Contrastive Learning of Visual Representations, ICML 2020
> >
> > [3] Unsupervised Learning of Visual Features by Contrasting Cluster Assignments, NeurIPS 2020
> >
> > [4] ConVIRT: Contrastive Learning of Medical Visual Representations from Paired Images and Text, Nature Biomedical Engineering 2021
> >
> > [5] RegionCLIP: Region-Based Language–Image Pretraining, CVPR 2023
> >
> > [6] Region-centric Image-Language Pretraining for Open-Vocabulary Detection, ECCV 2024

---

> > > ### Author Response · Authors · 2025-11-25
> > >
> > > **W3. Proposition 1 appears rather trivial. It is widely recognized that increasing the batch size can enhance the effectiveness of the InfoNCE loss.**
> > >
> > > **A.** We agree with the reviewer that the core phenomenon underlying Proposition 1 (i.e., that increasing batch size improves effectiveness of InfoNCE) is a well-known property of contrastive learning. Our intention was not to present Proposition 1 as a new theoretical contribution, but rather to use a simple formal statement to *clarify how this known property manifests in the specific setting of QueryREPA with modality-balanced batch sampling*.
> > >
> > > To avoid any misunderstanding, in the updated manuscript we relabel Proposition 1 as **Remark 1**, and explicitly state that "this remark restates a standard property of InfoNCE; we include it to clarify its implications under modality-balanced batch sampling in QueryREPA.”
> > >
> > > In particular, Proposition 1 is meant to serve as a compact formal tool to (i) reinterpret this known InfoNCE behavior in the context of modality batch construction, and (ii) explain why QueryREPA is especially sensitive to per-modality batch size and motivates our design of modality-balanced sampling. The conceptual contribution therefore lies in linking a standard InfoNCE property to the multimodal detection scenario and our query–modality alignment mechanism, rather than in the mathematical novelty of the statement itself.
> > >
> > > ---
> > >
> > > **Minor W1. The paper should include more recent and competitive baselines.**
> > >
> > > **A.** Thank you for the suggestion. First, we would like to clarify that the original version of the manuscript already includes several recent and competitive methods, such as **DiffusionDet [1]** and **Grounding DINO [2]**, in addition to strong DETR-style baselines. In our updated manuscript, we have further strengthened our experimental comparison by adding two recent detectors: **RTMDet [3]** and **Co-DETR [4]**. Our proposed method effectively **boosts DINO [5] performance to the point where it surpasses originally stronger baselines**, demonstrating the practical value of (MoCA+QueryREPA) as a strong, modular upgrade for standard detection backbones.
> > >
> > >
> > > **References**
> > >
> > > [1] DiffusionDet: Diffusion Model for Object Detection, ICCV 2023
> > >
> > > [2] Grounding DINO: Marrying DINO with Grounded Pre-Training for Open-Set Object Detection, ECCV 2024
> > >
> > > [3] RTMDet: An Empirical Study of Designing Real-Time Object Detectors
> > >
> > > [4] DETRs with Collaborative Hybrid Assignments Training, ICCV 2023
> > >
> > > [5] DINO: DETR with Improved DeNoising Anchor Boxes for End-to-End Object Detection, ICLR 2023
> > >
> > > ---
> > >
> > > **Minor W2. It seems that each modality uses a separate, dedicated image encoder. The authors should provide clearer clarification on this point.**
> > >
> > > **A.** We apologize for the confusion. We clarify that each modality does not use a separate image encoder, and a single common image encoder is used for all modalities. As mentioned in the paper, the commonly used ResNet-50 image encoder is leveraged as the image encoder.
> > >
> > > ---
> > >
> > > **Minor W3. Suggestion to cite the paper “Multimodal Medical Image Binding via Shared Text Embeddings”.**
> > >
> > > **A.** Thank you for the suggestion. Indeed, the suggested paper shares a common motivation of covering diverse imaging modalities, though our work is more focused on the alignment of object queries for detection tasks. We have included an explanation of the suggested paper in our updated manuscript.

---

> > > > ### Comment · Reviewer_osNV · 2025-11-27
> > > >
> > > > The author's reply answered some of my questions. QueryREPA is a good idea, but the description in the manuscript is not clear enough. For example, it doesn't explain how the query embedding is obtained, and the figure presentation is also poor. If the author can improve the writing and illustrations, I will raise the score to 6.

---

> > > > > ### Author Response · Authors · 2025-11-27
> > > > >
> > > > > We sincerely thank the reviewer for the positive assessment of QueryREPA and for suggesting improvement of the description and the figure. To directly address these concerns we have revised and added the following:
> > > > >
> > > > >
> > > > > **Major improvement of Figure 1.** We have redesigned Figure 1 and rewritten its caption to clearly visualize the overall pipeline of QueryREPA.
> > > > >
> > > > > **Clarification of QueryREPA (including how query embeddings are obtained) in Sec. 3.3.** We have thoroughly revised Section 3.3 to clearly explain the query alignment process.
> > > > >
> > > > > **Explicit algorithmic description in Appendix I.** To further improve reproducibility and technical transparency, we added Algorithm 1 in Appendix I, which provides a complete PyTorch-style pseudocode of the QueryREPA pretraining procedure.
> > > > >
> > > > > We believe these changes resolve the clarity issues raised by the reviewer and greatly improve the readability and interpretability of QueryREPA. We sincerely appreciate the reviewer’s willingness to reconsider the score based on these revisions.

---

> > > > > > ### Comment · Reviewer_osNV · 2025-11-28
> > > > > >
> > > > > > Thanks for your reply. I have raised my score to 6.

---

> > > > > > > ### Author Response · Authors · 2025-11-30
> > > > > > >
> > > > > > > We sincerely thank the reviewer for the positive reassessment and for raising the score to 6. We truly appreciate the constructive feedback, which helped us significantly improve the clarity and presentation of our work.

---

### Official Review · Reviewer_u3on · 2025-11-01

**Soundness:** 4
**Presentation:** 4
**Contribution:** 3
**Rating:** 4
**Confidence:** 3

**Summary:**

The motivation of this paper is to generate representations that generalize across a mixed training corpus of multi-modality medical data using ideas from representational alignment. They focus on object detection problems and mix modality tokens in with object query tokens to allow the attention mechanism to vary by modality.

**Strengths:**

This alignment strategy can work with different types of encoders, providing flexibility.

Clear performance gains from using this method compared to and on top of related work. Shows that improvement is not dependent on specific implementation details unrelated to the proposed method .

Minimal computational overhead by design.

**Weaknesses:**

QueryREPA adds pretraining before end-to-end detection training, but this is a noted limitation.

The modality tokens are produced via a fixed prompt and encoded with off-the-shelf general or biomedical CLIP variants. This approach captures coarse modality and class semantics but may lack granularity for subtle medical image distinctions. The study does not analyze failure cases for rare or complex conditions and doesn't examine potential limits of using static text encoders for domains with volatile or ambiguous language (pathology subtypes, COVID lesions, etc.). Are better prompts available? How does tokenization performance degrade as class granularity increases or when new modalities are added?

Needs better evaluation against prior work involving representation alignment applied to similar problem settings (either natural or medical data domains), or more thorough explanation in the related work section of why prior works are not comparable.

**Questions:**

See previous sections.

---

> ### Author Response · Authors · 2025-11-25
>
> **W1. QueryREPA adds pretraining before end-to-end detection training, but this is a noted limitation.**
>
> **A**. While QueryREPA introduces an additional pretraining stage prior to end-to-end detection training, this step offers a practical advantage in real-world medical settings where bounding box annotations are scarce. QueryREPA performs *representation alignment at the query level*, requiring only image-level class labels and **no bounding-box supervision during the pretraining stage**. This makes it particularly suitable for large medical datasets in which detailed annotations are limited.
>
> To demonstrate the potential of QueryREPA, we show below how pretraining on extra images *without* bounding box labels can increase the final detection performance. As an example, we additionally leverage the widely used **NIH Chest X-ray** dataset, which contains over 112,000 images but **provides bounding-box annotations for fewer than 1,000 cases**. This leaves the vast majority of images unusable for conventional detection pretraining. In the table below, we compare between different datasets used for QueryREPA, while keeping the datasets used for detection training the same.
>
> | QueryREPA Dataset | Detection Training Dataset | Total AP | Total AP₅₀ | CXR AP | CXR AP₅₀ | MRI AP | MRI AP₅₀ | Colonoscopy AP | Colonoscopy AP₅₀ | Pathology AP | Pathology AP₅₀ | CT AP | CT AP₅₀ |
> |:------------------:|:----------------------------:|:---------:|:-----------:|:-------:|:---------:|:-------:|:---------:|:----------------:|:-----------------:|:-------------:|:---------------:|:------:|:--------:|
> | ✗                | (2) + (3)                  | 38.5     | 62.0       | 19.1   | 41.5     | 77.4   | 96.8     | 68.7            | 86.8             | 53.4         | 83.0           | 34.2  | 66.2    |
> | (2) + (3)        | (2) + (3)                  | 39.1     | 62.9       | 19.2   | 42.5     | 77.6   | 96.6     | 70.4            | 85.7             | **55.7**     | **85.9**       | **34.6** | 67.7  |
> | (1) + (2) + (3)  | (2) + (3)                  | **39.4** | **63.3**   | **19.6** | **42.6** | **78.0** | **97.1** | **72.6**        | **89.9**         | 54.7         | 84.8           | 34.1  | **68.0** |
>
>
> Specifically, the datasets are: (1) NIH with only class labels (2) NIH with bounding box labels (3) VinBigData with bounding box labels. These results are also included in **Table 4** of our updated manuscript.
>
> In the experiment above, we observe that **including images without bounding-boxes during QueryREPA pretraining provides measurable performance gains**. Thus, although QueryREPA adds an explicit pretraining step, it *mitigates annotation scarcity* by turning otherwise underutilized, weakly labeled medical data into a beneficial resource that improves detection performance. This provides an entirely new research avenue for performing weakly supervised or unsupervised learning in detection models. Whereas existing unsupervised learning methods have largely focused on *image features*, our approach instead opens an intriguing path toward unsupervised learning directly on *object queries*.

---

> > ### Author Response · Authors · 2025-11-25
> >
> > **W2. The study does not analyze failure cases for rare or complex conditions and doesn't examine potential limits of using static text encoders for domains with volatile or ambiguous language. Are better prompts available? How does tokenization performance degrade as class granularity increases or when new modalities are added?**
> >
> > **A**. We thank the reviewer for the intriguing comment regarding the design of modality tokens and the use of frozen text encoders. The reviewer makes a number of points in this comment, and below we handle each point in detail.
> >
> > **Prompt Design and semantic granularity.** In the proposed method, we choose a compact {CLASS in MODALITY} template for constructing text prompts (e.g., “Nodule in lung CT”, “Brain tumor in MRI”), so that each token explicitly encodes both the visual category and its imaging domain. To assess the suitability of this text design, we have added an explicit ablation over prompt templates in our updated manuscript (see **Appendix E** and **Table 8**). The ablation results are as below:
> >
> > | Text                    |   AP  | AP₅₀ | AP₇₅ | APₛ  | APₘ  | APₗ  |
> > |:-------------------------:|:------:|:------:|:------:|:------:|:------:|:------:|
> > | ✗                       | 37.7 | 58.6 | 37.8 | 19.9 | 29.8 | 41.7 |
> > | {MODALITY}              | 40.7 | 65.1 | 42.7 | 26.1 | 33.0 | 44.2 |
> > | {CLASS}                 | 41.0 | 65.4 | 43.2 | **27.6** | 33.6 | 44.5 |
> > | {CLASS in MODALITY} | **41.1** | **65.5** | **43.4** | 25.4 | **34.6** | **44.9** |
> >
> >
> > Concretely, we compare three variants: {CLASS in MODALITY}, {CLASS} only, and {MODALITY} only. All three variants significantly improve AP over the detector baseline without any tokens. However, {CLASS in MODALITY} consistently achieves the best performance, {CLASS} is second-best, and {MODALITY} shows lowest performance of the three. This indicates that (i) the tokens indeed carry non-trivial semantic signal, since even very simple {CLASS} or {MODALITY} prompts provide measurable gains, and (ii) jointly encoding class and modality is empirically preferable to either component alone. In particular, the fact that {CLASS} > {MODALITY} suggests that the tokens capture fine-grained, class-specific distinctions beyond a coarse modality label, while the additional modality phrase further sharpens disentanglement. Thus, if the tokens were only encoding very rough modality semantics, then the {MODALITY}-only variant would be expected to perform similarly to {CLASS in MODALITY}, which is not what we observe.
> >
> > **Static text encoders and robustness.** Our goal in this work is not to propose a new text encoder, but to show that aligning object queries to *any* reasonable set of text-derived anchors is already sufficient to improve multimodality detection. To that end, we deliberately use frozen, off-the-shelf encoders (CLIP, BiomedCLIP, PubMedCLIP) and show that MoCA + QueryREPA yields consistent gains across all three. This suggests that our improvements are driven primarily by the *architecture-level* representation alignment, rather than a particular choice of encoder or prompt. Future research could explore specialized medical language models or richer prompt templates. Regarding robustness, new modalities can be incorporated by simply adding a small number of additional {CLASS in MODALITY} prompts for that modality and running the same alignment procedure; the model architecture and training recipe do not change.
> >
> > **Rare or complex conditions.** We agree with the reviewer that cases for rare or complex conditions are an important direction. Our current datasets, however, already include several ambiguous or complex abnormalities (e.g., COVID-19 infection in lung CT, Non-neoplastic polyp in colon endoscope). Within this scope, MoCA + QueryREPA consistently improves overall and per-modality AP and qualitatively recovers missed or imprecise detections for small lesions across modalities (see **Figures 3 and 7-11**). A systematic study of extremely fine-grained pathological subtypes is a promising avenue for future work.

---

> > > ### Author Response · Authors · 2025-11-25
> > >
> > > **W3. Needs better evaluation against prior work involving representation alignment applied to similar problem settings (either natural or medical data domains), or more thorough explanation in the related work section of why prior works are not comparable.**
> > >
> > > **A**. We thank the reviewer for raising this point. We have updated the **related works section** of our manuscript to give a more thorough explanation about existing representation alignment methods and explain their core differences to our method. To clarify, existing representation alignment methods for object detection can be roughly grouped into three paradigms: **(i) backbone-level alignment** of global image features, where contrastive or CLIP-style objectives act only on outputs of the *image encoder* and do not modify the logic of the detection model at all (e.g., MoCo [1], SimCLR [2], SwAV [3], ConVIRT [4]); **(ii) region-centric alignment**, which aligns *local* pre-defined region or proposal features to text descriptions (e.g., RegionCLIP [5], DITO [6]); and **(iii) query-level alignment (ours)**, which directly aligns object-level embeddings or “queries” to compact modality tokens inside the decoder. Therefore, our framework differs from existing works in both what is being aligned *and* the target of the alignment. Thus, existing representation alignment methods are largely *complementary* to our method: in principle, it would be possible to pretrain the image encoder with backbone-level alignment and afterwards apply our query-level alignment.
> > >
> > > We instead evaluate against the more comparable family of detectors that inject text into the decoder, comparing against language-guided baselines such as GLIP [7] and Grounding DINO [8], which fuse text via cross-attention. Our method (MoCA + QueryREPA) boosts DINO [9] to effectively outperform such baselines, indicating that **query-level alignment to compact modality tokens** is more effective in this medical multimodality regime than standard cross-attention fusion from long text sequences.
> > >
> > > **References**
> > >
> > > [1] MoCo: Momentum Contrast for Unsupervised Visual Representation Learning, CVPR 2020
> > >
> > > [2] SimCLR: A Simple Framework for Contrastive Learning of Visual Representations, ICML 2020
> > >
> > > [3] Unsupervised Learning of Visual Features by Contrasting Cluster Assignments, NeurIPS 2020
> > >
> > > [4] ConVIRT: Contrastive Learning of Medical Visual Representations from Paired Images and Text, Nature Biomedical Engineering 2021
> > >
> > > [5] RegionCLIP: Region-Based Language–Image Pretraining, CVPR 2023
> > >
> > > [6] Region-centric Image-Language Pretraining for Open-Vocabulary Detection, ECCV 2024
> > >
> > > [7] GLIP: Grounded Language-Image Pre-training for Open-Vocabulary Object Detection, CVPR 2022
> > >
> > > [8] Grounding DINO: Marrying DINO with Grounded Pre-Training for Open-Set Object Detection, ECCV 2024
> > >
> > > [9] DINO: DETR with Improved DeNoising Anchor Boxes for End-to-End Object Detection, ICLR 2023

---

### Author Response · Authors · 2025-11-26
**General Response to All Reviewers**

We are deeply grateful to all Reviewers for their time, careful evaluation, and thoughtful feedback (Reviewers u3on, osNV, MW9v, pyJE). We appreciate that the Reviewers recognized the practical relevance of our framework for multimodality medical object detection, its broad compatibility with existing detectors, and its lightweight computational overhead.

Below, we briefly summarize the main revisions and additions in our updated manuscript. Updated or newly added text is highlighted in blue. Detailed, point-by-point responses are provided as separate comments to each review.

**[Revision] Section 2.1 lines 107 - 140**: added discussion of the recent M$^3$Bind (Liu et al., 2025) framework and explained how it shares a common motivation with our work.

**[Revision] Figure 1**: improved illustration and clarified explanation of QueryREPA.

**[Revision] Section 2.3 lines 161 - 172**: clarified how existing representation alignment methods can be roughly grouped into three paradigms, and explained the core differences of our query-level alignment compared to backbone- and region-level alignment.

**[Revision] Section 3.3 lines 262 - 283**: clarified explanation of QueryREPA.

**[Revision] Section 3.3 lines 306 - 317**: toned down “Proposition” to “Remark”, and explained that the InfoNCE property is included to clarify its implications under modality-balanced batch sampling in QueryREPA.

**[Addition] Table 1 lines 390 - 391**: added recent baselines (RTMDet, Co-DETR).

**[Addition] Section 4.3 and Table 4:** added large-scale NIH Chest X-ray experiments using weakly labeled (class-only) data for QueryREPA pretraining.

**[Addition] Section 4.4 and Table 5:** added ablation isolating the individual and joint effects of MoCA and QueryREPA.

**[Addition] Appendix E and Table 8:** added ablation over prompt templates ({MODALITY}, {CLASS}, {CLASS in MODALITY}) and explained its implications.

**[Addition] Appendix F and Table 9:** added results of integrating MoCA + QueryREPA into DiffusionDet and explained its implications.

**[Addition] Appendix G and Figure 6:** explained how our method addresses heterogeneous image statistics, and added UMAP visualizations and silhouette-score analysis of query representations before and after QueryREPA.

**[Addition] Appendix I, Algorithm 1:** added a pseudocode of the QueryREPA pretraining procedure for improved clarity and reproducibility.

---

### Meta-Review · Area_Chair_6Tq5 · 2026-01-08

**Summary:**

The paper introduces a lightweight representation alignment framework for multimodality medical object detection and reports consistent gains. Reviews were borderline: strengths are practicality and empirical improvements, while key decision factors were (1) borderline novelty, (2) baseline and ablation completeness, (3) whether evidence truly supports “alignment” rather than incremental engineering, (4) over-strong generality claims, (5) limited failure-mode/generalization analysis.

**Reviewer Concerns:**

Addressed: (1) clearer method and reproducibility; (2) stronger baseline coverage; (3) more complete ablations; (4) added analyses supporting representation alignment; (5) broader validation.



Outstanding: (1) novelty still limited; (2) “detector-agnostic” (generality) claims remain somewhat overstated; (3) limited failure-mode and cross-domaingeneralization analysis; (4) positioning vs large VLM-based alternatives could be sharper.

**Reviewer Scores:**

- u3on (4): likely 4 to 5.
- osNV (4): likely 4 to 6.
- pyJE (4): likely stay 4.
- MW9v (4): likely stay 4.

---

### Decision · Program_Chairs · 2026-01-26

Reject